# INFORMATION MAXIMIZATION AUTO-ENCODING

## ABSTRACT

We propose the Information Maximization Autoencoder (IMAE), an information theoretic approach to simultaneously learn continuous and discrete representations in an unsupervised setting. Unlike the Variational Autoencoder framework, IMAE starts from a stochastic encoder that seeks to map each input data to a hybrid discrete and continuous representation with the objective of maximizing the mutual information between the data and their representations. A decoder is included to approximate the posterior distribution of the data given their representations, where a high fidelity approximation can be achieved by leveraging the informative representations. We show that the proposed objective is theoretically valid and provides a principled framework for understanding the tradeoffs regarding informativeness of each representation factor, disentanglement of representations, and decoding quality.

## 1 INTRODUCTION

A central tenet for designing and learning a model for data is that the resulting representation should be compact yet informative. Therefore, the goal of learning can be formulated as finding informative representations about the data under proper constraints. Generative latent variable models are a popular approach to this problem, where a model parameterized by $\theta$ of the form $p_\theta(\boldsymbol{x}) = \int p_\theta(\boldsymbol{x}|\boldsymbol{z})p(\boldsymbol{z})d\boldsymbol{z}$ is used to represent the relationship between the data $\boldsymbol{x}$ and the low dimensional latent variable $\boldsymbol{z}$. The model is optimized by fitting the generative data distribution $p_\theta(\boldsymbol{x})$ to the training data distribution $\widehat{p}(\boldsymbol{x})$, which involves maximizing the likelihood for $\theta$. Typically, this model is intractable even for moderately complicated functions $p_\theta(\boldsymbol{x}|\boldsymbol{z})$ with continuous $\boldsymbol{z}$. To remedy this issue, variational autoencoder (VAE) (Kingma and Welling, 2013; Rezende et al., 2014) proposes to maximize the evidence lower bound (ELBO) of the marginal likelihood objective.

However, as was initially pointed out in (Hoffman and Johnson, 2016), maximizing ELBO also penalizes the mutual information between data and their representations. This in turn makes the representation learning even harder. Many recent efforts have focused on resolving this problem by revising ELBO. Generally speaking, these works fall into two lines. One of them targets "disentangled representations" by encouraging the statistical independence between representation components (Higgins et al., 2016; Kim and Mnih, 2018; Gao et al., 2018; Chen et al., 2018; Esmaeili et al., 2018), while the other line of work seeks to control or encourage the mutual information between data and their representations (Mary Phuong, 2018; Burgess et al., 2018; Alemi et al., 2017; Dupont, 2018; Zhao et al., 2017). However, these approaches either result in an invalid lower bound for the VAE objective or cannot avoid sacrificing the mutual information.

Instead of building upon the generative latent variable model, we start with a stochastic encoder $p_\theta(\boldsymbol{z}|\boldsymbol{x})$ and aim at maximizing the mutual information between the data $\boldsymbol{x}$ and its representations $\boldsymbol{z}$. In this setting, a reconstruction or generating phase can be obtained as the variational inference of the true posterior $p_\theta(\boldsymbol{x}|\boldsymbol{z})$. By explicitly seeking for informative representations, the proposed model yields better decoding quality. Moreover, we show that the information maximization objective naturally induces a balance between the informativeness of each latent factor and the statistical independence between them, which gives a more principled way to learn semantically meaningful representations without invalidating ELBO or removing individual terms from it.

Another contribution of this work is proposing a framework for simultaneously learning continuous and discrete representations for categorical data. Categorical data are ubiquitous in real-world tasks, where using a hybrid discrete and continuous representation to capture both categorical information

and continuous variation in data is more consistent with the natural generation process. In this work, we focus on categorical data that are similar in nature, *i.e.,* where different categories still share similar variations (features). We seek to learn semantically meaningful discrete representations while maintaining disentanglement of the continuous representations that capture the variations shared across categories. We show that, compared to the VAE based approaches, our proposed objective gives a more natural yet effective way for learning these hybrid representations.

## 2    RELATED WORK

Recently, there has been a surge of interest in learning interpretable representations. Among them, $\beta$-VAE (Higgins et al., 2016) is a popular method for learning disentangled representations, which modifies ELBO by increasing the penalty on the KL divergence between the variational posterior and the factorized prior. However, by using large weight for the KL divergence term, $\beta$-VAE also penalizes the mutual information between the data and the latent representations more than a standard VAE does, resulting in more severe under utilization of the latent representation space.

Several follow up works propose different approaches to address the limitations of $\beta$-VAE. (Dupont, 2018; Alemi et al., 2017; Burgess et al., 2018; Mary Phuong, 2018) propose to constrain the mutual information between the representations and the data by pushing its upper bound, *i.e.,* the KL divergence term in ELBO, towards a progressively increased target value. However, specifying and tuning this target value can itself be very challenging, which makes this method less practical. Moreover, this extra constraint results in an invalid lower bound for the VAE objective. Alternatively, (Zhao et al., 2017) drops the mutual information term in ELBO. By pushing only the aggregated posterior towards a factorial prior, they implicitly encourage independence across the dimensions of latent representations without sacrificing the informativeness of the representations. However, simply removing the mutual information term also violates the lower bound of the VAE objective.

Another relevant line of work (Gao et al., 2018; Kim and Mnih, 2018; Chen et al., 2018; Esmaeili et al., 2018) seek to learn disentangled representations by explicitly encouraging statistical independence between latent factors. They all propose to minimize the *total correlation* term of the latent representations, either augmented as an extra term to ELBO or obtained by reinterpreting or re-weighting the terms in the VAE objective, as a way to encourage statistical independence between the representation components. In contrast, we show that our information maximization objective inherently contains the total correlation term while simultaneously seeking to maximize the informativeness of each representation factor.

In this paper, we introduce a different perspective to the growing body of the VAE based approaches for unsupervised representation learning. Starting by seeking informative representations for the data, we follow a more intuitive way to maximize the mutual information between the data and the representations. Moreover, we augment the continuous representation with a discrete one, which allows more flexibilities to model real world data that are generated from different categories. We invoke the information maximization principle (Linsker, 1988; Bell and Sejnowski, 1995) with proper constraints implied by the objective itself to avoid degenerate solutions. The proposed objective gives a theoretically elegant yet effective way to learn semantically meaningful representations.

## 3    INFORMATION MAXIMIZATION REPRESENTATION LEARNING

Given data $\boldsymbol{x} \in \mathbb{R}^d$, we consider learning a hybrid continuous-discrete representation, denoted respectively with variables $\boldsymbol{z} \in \mathbb{R}^{K_1}$ and $\boldsymbol{y} \in \{1, \ldots, K_2\}$, using a stochastic encoder parameterized by $\theta$, *i.e.,* $p_\theta(\boldsymbol{y}, \boldsymbol{z}|\boldsymbol{x})$. We seek to learn compact yet semantically meaningful representations in the sense that they should be low dimensional but informative enough about the data. A natural approach is to maximize the mutual information (Cover and Thomas, 2012) $I_\theta(\boldsymbol{x}; \boldsymbol{y}, \boldsymbol{z})$ between the data and its representations under the constraint $K_1, K_2 \ll d$. Here the mutual information between two random variables, *e.g.,* $\boldsymbol{x}$ and $\boldsymbol{z}$, is defined as $I_\theta(\boldsymbol{x}; \boldsymbol{z}) = H_\theta(\boldsymbol{z}) - H_\theta(\boldsymbol{z}|\boldsymbol{x})$, where $H_\theta(\boldsymbol{z}) = -\mathbb{E}_{p_\theta(\boldsymbol{z})}[\log p_\theta(\boldsymbol{z})]$ is the entropy of $\boldsymbol{z}$ and $H_\theta(\boldsymbol{z}|\boldsymbol{x}) = -\mathbb{E}_{p_\theta(\boldsymbol{x}, \boldsymbol{z})}[\log p_\theta(\boldsymbol{z}|\boldsymbol{x})]$ is the conditional entropy of $\boldsymbol{z}$ given $\boldsymbol{x}$. The mutual information can be interpreted as the decrease in uncertainty of one random variable given another random variable. In other words, it quantifies how much information one random variable has about the other.

A probabilistic decoder $q_\phi(\boldsymbol{x}|\boldsymbol{y}, \boldsymbol{z})$ is adopted to approximate the true posterior $p_\theta(\boldsymbol{x}|\boldsymbol{y}, \boldsymbol{z})$, which can be hard to estimate or even intractable. The dissimilarity between them is optimized by minimizing the KL divergence $D_{\mathrm{KL}}\left(p_\theta(\boldsymbol{x}|\boldsymbol{y}, \boldsymbol{z})||q_\phi(\boldsymbol{x}|\boldsymbol{y}, \boldsymbol{z})\right)$. In summary, IMAE considers the following,

$$\text{maximize}_{\theta,\phi} \quad \beta_0 \mathrm{I}_\theta(\boldsymbol{x}; \boldsymbol{y}, \boldsymbol{z}) - D_{\mathrm{KL}}\left(p_\theta(\boldsymbol{x}|\boldsymbol{y}, \boldsymbol{z})||q_\phi(\boldsymbol{x}|\boldsymbol{y}, \boldsymbol{z})\right) \ . \tag{1}$$

Given that $\mathrm{H}(x)$ is independent of the optimization procedure, we can show that optimizing (1) is equivalent to optimize the following[1],

$$\text{maximize}_{\theta,\phi} \quad \mathrm{I}_\theta(\boldsymbol{x}; \boldsymbol{y}, \boldsymbol{z}) + \mathbb{E}_{p_\theta(\boldsymbol{x},\boldsymbol{y},\boldsymbol{z})}\left[\log q_\phi(\boldsymbol{x}|\boldsymbol{y}, \boldsymbol{z})\right], \quad \beta = \beta_0 - 1 > 0 \ . \tag{2}$$

We set $\beta > 0$ to balance between maximizing the informativeness of latent representations and maintaining the decoding quality. The second term is often interpreted as the "reconstruction error" which can be optimized using the reparameterization tricks proposed by (Kingma and Welling, 2013) and (Jang et al., 2016) for continuous representation $\boldsymbol{z}$ and discrete representation $\boldsymbol{y}$ respectively. Now we introduce proper method to optimize the first term $\mathrm{I}_\theta(\boldsymbol{x}; \boldsymbol{y}, \boldsymbol{z})$ in (2).

## 3.1 Simultaneously seeking informativeness and disentanglement

We first show that $\mathrm{I}_\theta(\boldsymbol{x}; \boldsymbol{y}, \boldsymbol{z})$ inherently involves two keys terms that quantify the informativeness of each representation factor and the statistical dependence between these factors. Assuming the conditional distribution of the representation $(\boldsymbol{y}, \boldsymbol{z})$ given $\boldsymbol{x}$ is factorial, we also assume the marginal distribution of $\boldsymbol{y}$ and $\boldsymbol{z}$ are independent, *i.e.*, $p_\theta(\boldsymbol{y}, \boldsymbol{z}) = p_\theta(\boldsymbol{y})p_\theta(\boldsymbol{z})$, then[1]

$$\mathrm{I}_\theta(\boldsymbol{x}; \boldsymbol{y}, \boldsymbol{z}) = \mathrm{I}_\theta(\boldsymbol{x}; \boldsymbol{y}) + \sum_{k=1}^{K_1} \mathrm{I}_\theta(\boldsymbol{x}; \boldsymbol{z}_k) - D_{\mathrm{KL}}\left(p_\theta(\boldsymbol{z})\big|\big|\Pi_{k=1}^{K_1}p_\theta(\boldsymbol{z}_k)\right) \ . \tag{3}$$

The first two terms of the RHS quantify how much information each latent factor, *i.e.*, $\boldsymbol{y}$ or $\boldsymbol{z}_k$, carry about the data. The last term is known as the *total correlation* of $\boldsymbol{z}$ (Watanabe, 1960), which quantifies the statistical independence between the continuous latent factors and achieves the minimum if and only if they are independent of each other.

As is implied by (3), maximizing $\mathrm{I}_\theta(\boldsymbol{x}; \boldsymbol{y}, \boldsymbol{z})$ can be conducted by maximizing informativeness of each latent factor while simultaneously promoting statistical independence between the continuous factors. Various Monte Carlo based sampling strategies have been proposed to optimize the total correlation term (Chen et al., 2018; Esmaeili et al., 2018); in this work we follow this line (see Appendix B). Next we proceed by constructing tractable approximations for $\mathrm{I}_\theta(\boldsymbol{x}; \boldsymbol{z}_k)$ and $\mathrm{I}_\theta(\boldsymbol{x}; \boldsymbol{y})$ respectively.

## 3.2 Informative continuous representations

Without any constraints, the mutual information $\mathrm{I}_\theta(\boldsymbol{x}; \boldsymbol{z}_k)$ between a continuous latent factor and data can be trivially maximized by severely fragmenting the latent space. To be more precise, consider the following proposition. While similar results have likely been established in the information theory literature, we include this proposition to motivate our objective design.

**Proposition 1.** *Suppose the conditional distribution $p_\theta(\boldsymbol{z}|\boldsymbol{x})$ is a factorial Gaussian distribution with mean $\mu(\boldsymbol{x})$ and covariance $\Sigma(\boldsymbol{x})$. Let $\sigma(\boldsymbol{x}) \in \mathbb{R}^{K_1}$ denote the diagonal entries of $\Sigma(\boldsymbol{x})$, then*

$$I_\theta(\boldsymbol{x}; \boldsymbol{z}_k) \le \frac{1}{2}\log\left[\left(\mathbb{E}_{\boldsymbol{x}}\left[\sigma_k^2(\boldsymbol{x})\right] + \mathrm{Var}_{\boldsymbol{x}}\left[\mu_k(\boldsymbol{x})\right]\right)\right] - \frac{1}{2}\mathbb{E}_{\boldsymbol{x}}\left[\log\sigma_k^2(\boldsymbol{x})\right] \ , \quad k = 1, \ldots, K_1 \ . \tag{4}$$

*The equality in (4) is attained if and only if $\boldsymbol{z}_k$ is Gaussian distributed, given which we have*

$$I_\theta(\boldsymbol{x}; \boldsymbol{z}_k) \ge \frac{1}{2}\log\left(1 + \mathrm{Var}_{\boldsymbol{x}}\left[\mu_k(\boldsymbol{x})\right]/\mathbb{E}_{\boldsymbol{x}}\left[\sigma_k^2(\boldsymbol{x})\right]\right) \ , \quad k = 1, \ldots, K_1 \ . \tag{5}$$

Note here both $\mu_k(\boldsymbol{x})$ and $\sigma_k(\boldsymbol{x})$ are random variables. The above result implies that $\boldsymbol{z}_k$ is more informative about $\boldsymbol{x}$ if it has less uncertainty given $\boldsymbol{x}$ yet captures more variance in data, *i.e.*, $\sigma_k(\boldsymbol{x})$ is small while $\mu_k(\boldsymbol{x})$ disperses within a large space. However, this can result in discontinuity of $\boldsymbol{z}_k$, where in the extreme case each data sample is associated with a delta distribution in the latent space.

---

[1] Detailed derivation is provided in Appendix A.

In light of this, we can make what we described above more precise. A vanishing variance of the conditional distribution $p(z_k|x)$ leads to a plain autoencoder that maps each data sample to a deterministic latent point, which can fragment the latent space in a way that each data sample corresponds with a delta distribution in the latent space $p_\theta(z_k|x^{(i)}) = \delta(z_k^{(i)})$. On the other hand, Proposition 1 also implies that controlling the variance $\sigma_k(x)$ to be finite, $I_\theta(x; z_k)$ will be maximized by pushing $\mu_k(x)$ towards two extremes ($\pm\infty$). To remedy this issue while achieving the upper bound, a natural resolution is to squeeze $z_k$ within the domain of a Gaussian distribution with finite mean and variance. By doing so, we can avoid the degenerate solution while achieving a more reasonable trade-off between enlarging the spread of $\mu_k(x)$ and maintaining the continuity of $z$. Therefore, we consider the following as the surrogate for maximizing $I_\theta(x; z_k)$,

$$\text{maximize } \mathcal{L}_\theta(z) := -\sum_{k=1}^{K_1} D_{\text{KL}}\left(p_\theta(z_k)||r(z_k)\right) . \tag{6}$$

Here $r(z_k)$ are i.i.d scaled normal distribution with finite variance. That is, we push each $p_\theta(z_k)$ towards a Gaussian distribution $r(z_k)$ by minimizing the KL divergence between them.

## 3.3 INFORMATIVE DISCRETE REPRESENTATIONS

Unlike the continuous representation, the mutual information $I_\theta(x; y)$ between a discrete representation and data can be well approximated, given the fact that the cardinality of the space of $y$ is typically low. To be more specific, given $N$ i.i.d samples $\{x_n\}_{n=1}^N$ of the data, the empirical estimation of $I_\theta(x; y)$ under the conditional distribution $p_\theta(y|x_n)$ follows as

$$\widehat{I}_\theta(x; y) = \widehat{H}_\theta(y) - \widehat{H}_\theta(y|x) = H\left(\frac{1}{N}\sum_{n=1}^N p_\theta(y|x_n)\right) - \frac{1}{N}\sum_{n=1}^N H\left(p_\theta(y|x_n)\right) . \tag{7}$$

As shown in Proposition 2, with a suitably large batch of samples, the empirical mutual information $\widehat{I}_\theta(x; y)$ is a good approximation to $I_\theta(x; y)$. This enables us to optimize $I_\theta(x; y)$ in a theoretically justifiable way that is amenable to stochastic gradient descent with minibatches of data.

**Proposition 2.** *Let $y$ be a discrete random variable that belongs to some categorical class $\mathcal{C}$. Assume the marginal probabilities of the true and the predicted labels are bounded below, i.e. $p_\theta(y), \widehat{p}_\theta(y) \in [1/(CK_2), 1]$ for all $y \in \mathcal{C}$ with some constant $C > 1$. Then for any $\delta \in (0, 1)$,*

$$\mathbb{P}\left(\left|I_\theta(x; y) - \widehat{I}_\theta(x; y)\right| \leq K_2\left(\max\{\log CK_2 - 1, 1\} + e\right)\sqrt{\frac{\log(2K_2/\delta)}{2N}}\right) \geq 1 - 2\delta . \tag{8}$$

*Here $N$ denotes the number of samples used to establish $\widehat{I}_\theta(x; y)$ according to Eq (7).*

Therefore, to maximize the mutual information $I_\theta(x; y)$, we consider the following:

$$\max \ \mathcal{L}_\theta(y) := \widehat{I}_\theta(x; y). \tag{9}$$

Maximizing the the mutual information $I_\theta(x; y)$ provides a natural way to learn discrete categorical representations. To see this, notice that $I_\theta(x; y)$ contains two fundamental quantities, the category balance term $H_\theta(y)$ and the category separation term $H_\theta(y|x)$. In other words, maximizing $I_\theta(x; y)$ trades off uniformly assigning data over categories and seeking highly confident categorical identity for each sample $x$. The maximum is achieved if $p_\theta(y|x)$ is deterministic while the marginal distribution $p_\theta(y)$ is uniform, that is $H_\theta(y|x) = 0$ and $H_\theta(y) = \log K_2$.

**Overall Objective** As a summary of (3) (6) and (9), our overall objective is

$$\beta\left(\max_{\theta,\phi} \ \mathcal{L}_\theta(z) + \mathcal{L}_\theta(y) - D_{\text{KL}}\left[p(z)||\Pi_{k=1}^{K_1}p(z_k)\right]\right) + \mathbb{E}_{p_\theta(x,y,z)}\left[\log q_\phi(x|y, z)\right] .$$

The first three terms associate with our information maximization objective, while the last one aims at better approximation of the posterior $p_\theta(x|y, z)$. A better balance between these two targets can be achieved by weighting them differently. One the other hand, the informativeness of each latent factor can be optimized through $\mathcal{L}_\theta(z)$ and $\mathcal{L}_\theta(y)$, while statistically independent latent continuous factors can be promoted by minimizing the total correlation term $D_{\text{KL}}\left[p(z)||\Pi_{k=1}^{K_1}p(z_k)\right]$.

Therefore, trade-offs can be formalized regarding the informativeness of each latent factor, disentanglement of the representation, and better decoding quality. This motivates us to consider the following objective, let $\beta, \gamma > 0$,

$$\max_{\theta,\phi} \ \mathcal{L}_{\text{IMAE}} := \mathbb{E}_{p_\theta(x,y,z)}\left[\log q_\phi(x|y, z)\right] + \beta\mathcal{L}_\theta(y) + \beta\mathcal{L}_\theta(z) - \gamma D_{\text{KL}}\left[p_\theta(z)||\Pi_{k=1}^{K_1}p_\theta(z_k)\right] . \tag{10}$$

## 4 EXPERIMENTAL RESULTS

We compare IMAE against various VAE based approaches that are summarized in Figure 1. We would like to demonstrate that IMAE can (i) successfully learn a hybrid of continuous and discrete representations, with $\boldsymbol{y}$ matching the intrinsic categorical information $\boldsymbol{y_{\text{true}}}$ well and $\boldsymbol{z}$ capturing the disentangled feature information shared across categories; (ii) outperform the VAE based models by achieving a better trade-off between representation interpretability and decoding quality. We choose the priors $r(\boldsymbol{z})$ and $r(\boldsymbol{y})$ to be the isotropic Gaussian distribution and uniform distribution respectively. Detailed experimental settings are provided in Appendix G.

$$\mathcal{L}_{\text{VAE}} = \boxed{\mathbb{E}_{p(\boldsymbol{y},\boldsymbol{z}|\boldsymbol{x})}\left[q(\boldsymbol{x}|\boldsymbol{y},\boldsymbol{z})\right] - D_{\text{KL}}\left(p(\boldsymbol{z}|\boldsymbol{x})||r(\boldsymbol{z})\right) - D_{\text{KL}}\left(p(\boldsymbol{y}|\boldsymbol{x})||r(\boldsymbol{y})\right)} \leftarrow \text{ELBO}$$

$$= \underbrace{\mathbb{E}_{p(\boldsymbol{y},\boldsymbol{z}|\boldsymbol{x})}\left[q(\boldsymbol{x}|\boldsymbol{y},\boldsymbol{z})\right]}_{\textcircled{1}} - \underbrace{I(\boldsymbol{x};\boldsymbol{y})}_{\textcircled{2}} - \underbrace{D_{\text{KL}}\left(p(\boldsymbol{y})||r(\boldsymbol{y})\right)}_{\textcircled{3}} - \underbrace{I(\boldsymbol{x};\boldsymbol{z})}_{\textcircled{4}} - \underbrace{D_{\text{KL}}\left(p(\boldsymbol{z})||r(\boldsymbol{z})\right)}_{\textcircled{5}}$$

$\beta$-VAE: $\textcircled{1} - \beta\left(\textcircled{2} + \textcircled{3}\right) - \beta\left(\textcircled{4} + \textcircled{5}\right)$     InfoVAE: $\textcircled{1} - \beta\,\textcircled{3} - \beta\,\textcircled{5}$

Joint-VAE: $\textcircled{1} - \beta\left|\textcircled{2} + \textcircled{3} - C_{\boldsymbol{y}}\right| - \beta\left|\textcircled{4} + \textcircled{5} - C_{\boldsymbol{z}}\right|$

Figure 1: Summarization of relevant work. $\beta$-VAE modifies ELBO by increasing the penalty on the KL divergence terms. InfoVAE drops the mutual information terms from ELBO. JointVAE seeks to control the mutual information by pushing the their upper bounds (the associated KL divergence terms) towards progressively increased values, $C_{\boldsymbol{y}}\&C_{\boldsymbol{z}}$. We drop the subscripts $\theta$ and $\phi$ hereafter.

### 4.1 INFORMATIVE REPRESENTATIONS YIELD BETTER INTERPRETABILITY

We first qualitatively demonstrate that informative representations can yield better interpretability. For the continuous representation, Figure 2 validates Proposition 1 by showing that, with roughly same amount of variance for each latent variable $\boldsymbol{z}_k$, those achieving high mutual information with the data have mean values $\mu_k(\boldsymbol{x})$ of the conditional probability $p(\boldsymbol{z}_k|\boldsymbol{x})$ disperse across data samples and variances $\sigma_k(\boldsymbol{x})$ decrease to small values for all data samples. As a qualitative evaluation, we traverse latent dimensions corresponding with different levels of $\text{I}(\boldsymbol{x},\boldsymbol{z}_k)$. As seen in Figure 2(b)-(d), informative variables in the continuous representation have uncovered intuitive continuous factors of the variation in the data, while the factor $\boldsymbol{z}_8$ has no mutual information with the data and shows no variation. We observe the same phenomenon for the discrete representation $\boldsymbol{y}$ in Figure 2(e)&(f), which were obtained with two different values of $\beta$ and $\gamma$, where the more informative one discovers matches the natural labels better. This provides further evidence for that interpretable latent factors can be attained by maximizing the mutual information between the representations and the data.

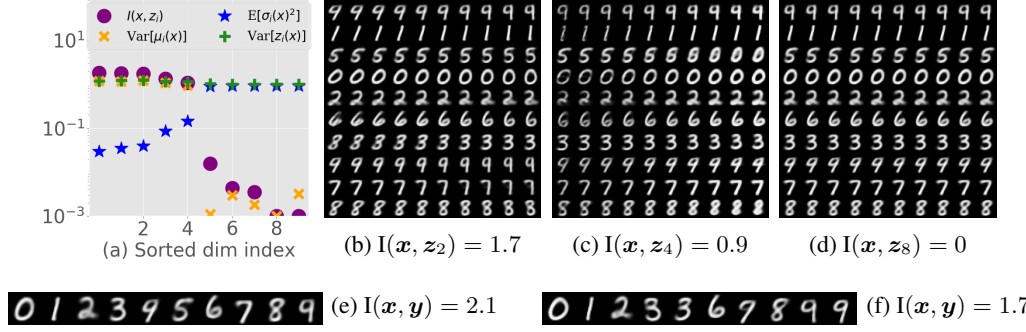

Figure 2: IMAE on MNIST **(a)** Illustration of Proposition 1. **(b)-(d)** Latent traverse on the continuous representations $\boldsymbol{z}$. The rows are conditioned on the discrete representations $\boldsymbol{y}$ learnt by IMAE, and the initial value of $\boldsymbol{z}$ for each row is obtained by feeding the encoder with randomly selected data corresponds with $\boldsymbol{y}$. We then manipulate each selected $\boldsymbol{z}_k$ within $[-2,2]$ while keeping all other dimensions fixed. **(e) & (f)** Discrete representations learnt by IMAE with different $\beta$ values.

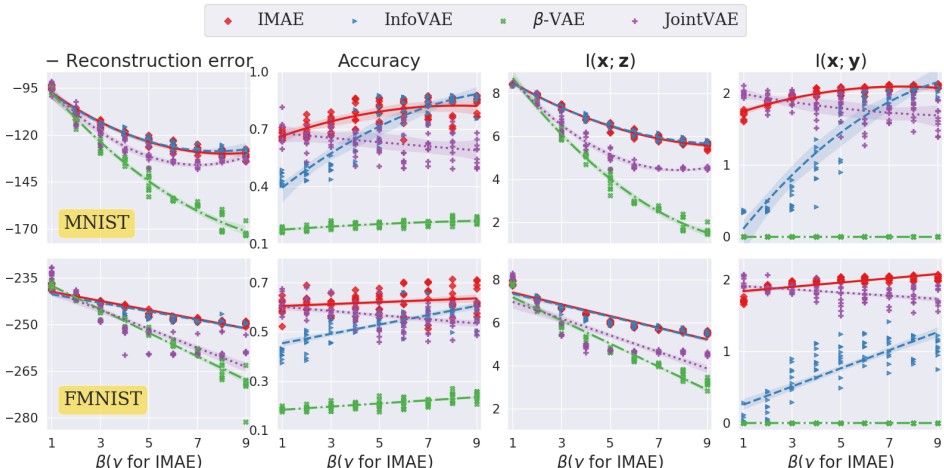

Figure 3: Tracking the key quantities for different models by sweeping $\beta$ for all different methods. We set $\gamma = 2\beta$ for IMAE. For each $\beta$, we run each method over 10 random initializations.

## 4.2 QUANTITATIVE COMPARISONS

In this section, we perform quantitative evaluations on MNIST (LeCun and Cortes, 2010), Fashion MNIST (Xiao et al., 2017) and dSprites (Matthey et al., 2017). We show that IMAE achieves better interpretability vs. decoding quality trade-off.

**Unsupervised learning of discrete latent factor**   Before we present our main results, we first describe an assumption that we make on the discrete representations. For the discrete representation, a reasonable assumption is that the conditional distribution $p(\boldsymbol{y}|\boldsymbol{x})$ should be locally smooth so that the data samples that are close on their manifold should have high probability of being assigned to the same category (Agakov, 2005). This assumption is crucial for using neural networks to learn discrete representations, since it's easy for a high capacity model to learn a non-smooth function $p(\boldsymbol{y}|\boldsymbol{x})$ that can abruptly change its predictions without guaranteeing similar data samples will be mapped to similar $\boldsymbol{y}$. To remedy this issue, we adopt the virtual adversarial training (VAT) trick proposed by (Miyato et al., 2016) and augment $\mathcal{L}_\theta(\boldsymbol{y})$ as follows:[2]

$$\max\ \mathcal{L}_\theta(\boldsymbol{y}) := \widehat{\mathrm{I}}_\theta(\boldsymbol{x};\boldsymbol{y}) - \mathbb{E}_{\widehat{p}(\boldsymbol{x})}\left[\max_{\|\eta\|\leq\epsilon}\mathrm{H}\left(p_\theta(\boldsymbol{y}|\boldsymbol{x}); p_\theta(\boldsymbol{y}|\boldsymbol{x}+\eta)\right)\right]\ . \tag{11}$$

The second term of RHS regularizes $p_\theta(\boldsymbol{y}|\boldsymbol{x})$ to be consistent within the $\epsilon$ norm ball of each data sample so as to maintain the local smoothness of the prediction model. *For fair comparison, we augment all four methods with VAT. As demonstrated in Appendix D, using VAT is essential for all of them except $\beta$-VAE to learn interpretable discrete representations.*

### 4.2.1 MNIST AND FASHION MNIST

We start by evaluating different methods on MNIST and Fashion MNIST, for which we train over a range of $\beta$ values (we set $\gamma = 2\beta$ for IMAE).

**Discrete representations**   For the discrete representations, by simply pushing the conditional distribution $p(\boldsymbol{y}|\boldsymbol{x})$ towards the uniform distribution $r(\boldsymbol{y})$, $\beta$-VAE sacrifices the mutual information $\mathrm{I}(\boldsymbol{x};\boldsymbol{y})$ and hence struggles in learning interpretable discrete representation even with VAT. As a comparison, InfoVAE performs much better by dropping $\mathrm{I}(\boldsymbol{x};\boldsymbol{y})$ from ELBO. For data that are distinctive enough between categories (MNIST), with large $\beta$ values InfoVAE performs well by uniformly distributing the whole data over categories through minimizing $D_{\mathrm{KL}}(p(\boldsymbol{y})\|r(\boldsymbol{y}))$ while simultaneously encouraging local smoothness with VAT. However, InfoVAE struggles with less distinctive data (Fashion-MNIST), where it cannot give fairly confident category separation by only

---

[2]In this paper, we set $\epsilon = 1$ across datasets. VAT can be effectively approximated by a pair of forward and backward passes (Miyato et al., 2016).

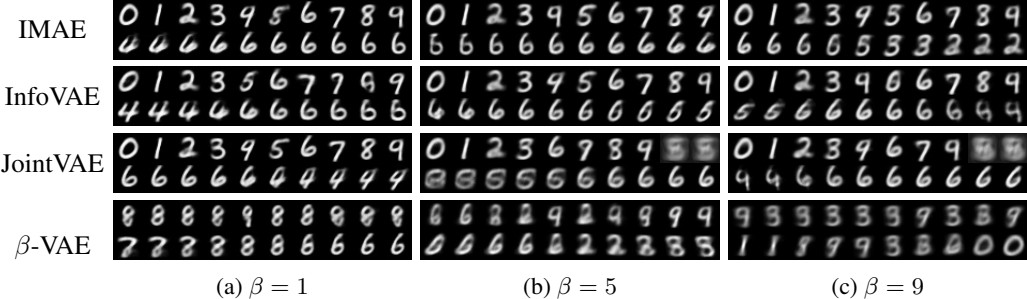

(a) $\beta = 1$     (b) $\beta = 5$     (c) $\beta = 9$

Figure 4: For each image, the first row is the digit type learnt by the model, where each entry is obtained by feeding the decoder with the averaged $z$ values corresponding with the learnt $y$. The second row is obtained by traversing the "angle" latent factor within $[-2, 2]$ on digit 6. IMAE is capable of uncovering the underlying discrete factor over a wide range of $\beta$ values. More interpretable continuous representations can be obtained when the method is capable of learning discrete representations, since less overlap between the mainfolds of each category is induced.

requiring local smoothness. In contrast, IMAE achieves much better performance by explicitly encouraging confident category separation via minimizing the conditional entropy $\mathrm{H}(y|x)$, while using VAT to maintain local smoothness so as to prevent overfitting of neural network. Although JointVAE performs much better than $\beta$-VAE by pushing the upper bound of $\mathrm{I}(x; y)$ towards a progressively increasing target value $C_y$, we found it can easily get stuck at some bad local optima where $\mathrm{I}(x; y)$ is comparatively large while the accuracy is poor. A heuristic is that once JointVAE enters the local region of a local optima, progressively increasing $C_y$ only induces oscillation within that region. [3]

**Informativeness, interpretability and decoding quality** As illustrated in Figure 1, by using large $\beta$ values, $\beta$-VAE sacrifices more mutual information between the data and its representations, which in turn (see Figure 3) results in less informative representations followed by poor decoding quality. In contrast, the other three methods can remedy this issue to different degrees, and hence attains better trade-off regarding informativeness of latent representations and decoding quality. Compared to JointVAE and InfoVAE, IMAE is more capable of learning discrete presentations over a wide range of $\beta, \gamma$ values, which implies less overlap between the manifolds of different categories is induced. As a result, IMAE is expected to yield better decoding quality for each category. Although InfoVAE and JointVAE can also learn comparatively good discrete representations when using large and small $\beta$ values respectively, the corresponding results of these two regions associate with either poor decoding quality or much lower disentanglement score (see section 4.2.2). In contrast, IMAE consistently performs well with different hyperparameters, especially in the region of interest where the decoding quality as well as the informativeness of latent representations are good enough.

### 4.2.2 2D SHAPES

In this section, we quantitatively evaluate the disentanglement capability of IMAE on dSprites where the ground truth factors of both continuous and discrete representaions are available. We use the disentanglement metric proposed by (Chen et al., 2018), which is defined in terms of the gap between the top two empirical mutual information of each latent representation factor and a ground truth factor. The disentanglement score is defined as the weighted average of the gaps. A high disentanglement score implies that each ground truth factor associates with one single representation factor that is more informative than the others, *i.e.,* the learnt representation factors are more disentangled.[4]

Figure 5 shows that, with large $\beta$ values, $\beta$-VAE penalizes the mutual information too much and this degrades the usefulness of representations. while all other three methods achieve higher disentanglement score with better decoding quality. For JointVAE, higher $\beta$ values push the upper bound of mutual information converges to the prefixed target value, it therefore can maintain more mutual

---

[3]More results of JointVAE can be found in Appendix F.

[4]Although the truth discrete factor is provided, we evaluate the disentanglement quality only in terms of the continuous representations since the pixel-wise difference between different categories are very small. The results of considering the disentanglement score regrading both $y$ and $z$ is provided in Appendix E.

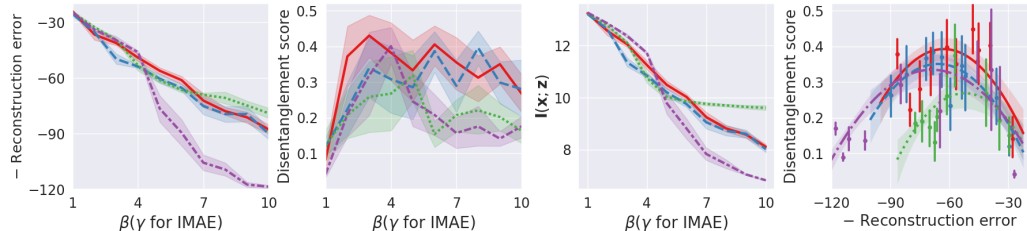

(a) IMAE performs well regarding the disentanglement score vs. decoding quality trade-off, especially in the region of interest where both decoding quality and informativeness of representations are fairly good.

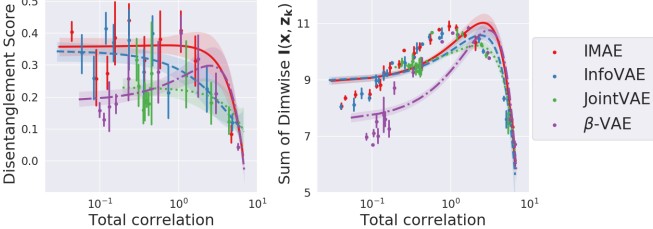

(b) Negative correlation between total correlation and disentanglement score. It also implies that the disentanglement score tends to decrease along with the total correlation if using even larger $\beta$, due to the diminishing informativeness of representation factors. In the extreme case, both total correlation and disentanglement score can degrade to zero.

Figure 5: **Disentanglement comparison on dSprites.** The results are reported by training each method with $\beta \in [1, 10]$, and we set $\beta = \gamma/2$ with $\gamma \in [1, 10]$ for IMAE. For each $\beta$ value, every method is trained over 8 random initializations. Shade regions indicate the $80\%$ confidence intervals.

information between the data and the whole latent representations and give better decoding quality. However, the disentanglement quality is poor in this region, which implies that simply restricting the overall capacity of the latent representations is not enough for learning disentangled representations. While InfoVAE yields comparatively better disentanglement score by pushing the marginal joint distribution of the representations towards a factorial distribution harder with large values of $\beta$, the associated decoding quality and informativeness of latent representations are both poor. In contrast, IMAE is capable of achieving better trade-off between the disentanglement score and the decoding quality in the region of interest where the decoding quality as well as the informativeness are fairly good. We attribute this to the effect of explicitly seeking for statistically independent latent factors by minimizing the total correlation term in our objective.

## 5 CONCLUSION

We have proposed IMAE, a novel approach for simultaneously learning the categorical information of data while uncovering latent continuous features shared across categories. Different from VAE, IMAE starts with a stochastic encoder that seeks to maximize the mutual information between data and their representations, where a decoder is used to approximate the true posterior distribution of the data given the representations. This model targets at informative representations directly, which in turn naturally yields an objective that is capable of simultaneously inducing semantically meaningful representations and maintaining good decoding quality, which is further demonstrated by the numerical results.

Unsupervised joint learning of disentangled continuous and discrete representations is a challenging problem due to the lack of prior for semantic awareness and other inherent difficulties that arise in learning discrete representations. This work takes a step towards achieving this goal. A limitation of our model is that it pursues disentanglement by assuming or trying to encourage independent scalar latent factors, which may not always be sufficient for representing the real data. For example, data may exhibit category specific variation, or a subset of latent factors might be correlated. This motivates us to explore more structured disentangled representations; one possible direction is to encourage group independence. We leave this for future work.

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

## A   PROOF OF SECTION 3

**Balance between posterior inference fidelity and information maximization**   Notice that we can rewrite the mutual information between the data $\boldsymbol{x}$ and its representations as the following,

$$I_\theta(\boldsymbol{x};\boldsymbol{y},\boldsymbol{z}) = \mathrm{H}(\boldsymbol{x}) + \mathbb{E}_{p_\theta(\boldsymbol{x},\boldsymbol{y},\boldsymbol{z})}\left[\log q_\phi(\boldsymbol{x}|\boldsymbol{y},\boldsymbol{z})\right] + D_{\mathrm{KL}}\left[p_\theta(\boldsymbol{x}|\boldsymbol{y},\boldsymbol{z})||q_\phi(\boldsymbol{x}|\boldsymbol{y},\boldsymbol{z})\right] \ . \tag{12}$$

It then follows that,

$$I_\theta(\boldsymbol{x};\boldsymbol{y},\boldsymbol{z}) - D_{\mathrm{KL}}\left(p_\theta(\boldsymbol{x}|\boldsymbol{y},\boldsymbol{z})||q_\phi(\boldsymbol{x}|\boldsymbol{y},\boldsymbol{z})\right) = \mathrm{H}(\boldsymbol{x}) + \mathbb{E}_{p_\theta(\boldsymbol{x},\boldsymbol{y},\boldsymbol{z})}\left[\log q_\phi(\boldsymbol{x}|\boldsymbol{y},\boldsymbol{z})\right] \tag{13}$$

Since $\mathrm{H}(\boldsymbol{x})$ is independent of the optimization procedure, we have the following,

$$\max \ \ \beta I_\theta(\boldsymbol{x};\boldsymbol{y},\boldsymbol{z}) - D_{\mathrm{KL}}\left(p_\theta(\boldsymbol{x}|\boldsymbol{y},\boldsymbol{z})||q_\phi(\boldsymbol{x}|\boldsymbol{y},\boldsymbol{z})\right), \qquad \beta > 1$$
$$\Rightarrow \max \ \ (\beta - 1)I_\theta(\boldsymbol{x};\boldsymbol{y},\boldsymbol{z}) + \mathbb{E}_{p_\theta(\boldsymbol{x},\boldsymbol{y},\boldsymbol{z})}\left[\log q_\phi(\boldsymbol{x}|\boldsymbol{y},\boldsymbol{z})\right] \tag{14}$$

where $\beta$ trade-off the informativeness of the latent representation and generation fidelity.

**Decomposition of $I_\theta(\boldsymbol{x};\boldsymbol{y},\boldsymbol{z})$**   Let $\boldsymbol{b} = (\boldsymbol{z},\boldsymbol{y})$ denote the joint random variable consisting of the continuous random variable $\boldsymbol{b}$ and discrete random variable $\boldsymbol{y}$.

Note that $I_\theta(\boldsymbol{x};\boldsymbol{y},\boldsymbol{z}) = I_\theta(\boldsymbol{x};\boldsymbol{b})$ can be written as:

$$\begin{aligned}
I_\theta(\boldsymbol{x};\boldsymbol{b}) &= -\int_\mathcal{X} p(\boldsymbol{x})\int_\mathcal{Z} p_\theta(\boldsymbol{b}|\boldsymbol{x})\log p_\theta(\boldsymbol{b})d\boldsymbol{b}d\boldsymbol{x} + \int_\mathcal{X} p(\boldsymbol{x})\int_\mathcal{Z} p_\theta(\boldsymbol{b}|\boldsymbol{x})\log p_\theta(\boldsymbol{b}|\boldsymbol{x})d\boldsymbol{b}d\boldsymbol{x}\\
&= -\int_\mathcal{Z} p_\theta(\boldsymbol{b})\log p_\theta(\boldsymbol{b})d\boldsymbol{b} + \int_\mathcal{X} p(\boldsymbol{x})\int_\mathcal{Z} p_\theta(\boldsymbol{b}|\boldsymbol{x})\log p_\theta(\boldsymbol{b}|\boldsymbol{x})d\boldsymbol{b}d\boldsymbol{x} \ . \tag{15}
\end{aligned}$$

The second term in Eq (15) has the form:

$$\begin{aligned}
\int_\mathcal{X} p(\boldsymbol{x})\int_\mathcal{Z} p_\theta(\boldsymbol{b}|\boldsymbol{x})\log p_\theta(\boldsymbol{b}|\boldsymbol{x})d\boldsymbol{b}d\boldsymbol{x} &\overset{\vartheta_1}{=} \sum_{k=1}^{K_1+1}\int_\mathcal{X} p(\boldsymbol{x})\int_\mathcal{Z} p_\theta(\boldsymbol{b}|\boldsymbol{x})\log p_\theta(\boldsymbol{b}_k|\boldsymbol{x})d\boldsymbol{b}d\boldsymbol{x}\\
&= \sum_{k=1}^{K_1+1} H_\theta(\boldsymbol{b}_k|\boldsymbol{x}) \ , \tag{16}
\end{aligned}$$

where $\vartheta_1$ follows by the assumption that $p_\theta(\boldsymbol{b}|\boldsymbol{x})$ is factorial.

For the first term in Eq (15), we have:

$$\begin{aligned}
\int_\mathcal{Z} p_\theta(\boldsymbol{b})\log p_\theta(\boldsymbol{b})d\boldsymbol{b} &= \int_\mathcal{Z} p_\theta(\boldsymbol{b})\log\frac{p_\theta(\boldsymbol{b})}{\Pi_{k=1}^{K_1+1}p_\theta(\boldsymbol{b}_k)}d\boldsymbol{b} + \sum_{k=1}^{K_1+1}\int_\mathcal{Z} p_\theta(\boldsymbol{b})\log p_\theta(\boldsymbol{b}_k)d\boldsymbol{b}\\
&= D_{\mathrm{KL}}\left(p_\theta(\boldsymbol{b})||\Pi_{k=1}^{K_1+1}p_\theta(\boldsymbol{b}_k)\right) - \sum_{k=1}^{K_1+1} H_\theta(\boldsymbol{b}_k) \ . \tag{17}
\end{aligned}$$

Substituting Eqs (16) & (17) into Eq (15) yields the result:

$$\begin{aligned}
I_\theta(\boldsymbol{x};\boldsymbol{y},\boldsymbol{z}) = I_\theta(\boldsymbol{x};\boldsymbol{b}) &= H_\theta(\boldsymbol{b}_k) - D_{\mathrm{KL}}\left(p_\theta(\boldsymbol{b})||\Pi_{k=1}^{K_1+1}p_\theta(\boldsymbol{b}_k)\right) - \sum_{k=1}^{K_1+1} H_\theta(\boldsymbol{b}_k|\boldsymbol{x})\\
&= \sum_{k=1}^{K_1+1} I_\theta(\boldsymbol{x};\boldsymbol{b}_k) - D_{\mathrm{KL}}\left(p_\theta(\boldsymbol{b})||\Pi_{k=1}^{K_1+1}p_\theta(\boldsymbol{b}_k)\right)\\
&= I_\theta(\boldsymbol{x};\boldsymbol{y}) + \sum_{k=1}^{K_1} I_\theta(\boldsymbol{x};\boldsymbol{z}_k) - D_{\mathrm{KL}}\left(p_\theta(\boldsymbol{y},\boldsymbol{z})||p_\theta(\boldsymbol{y})\Pi_{k=1}^{K_1}p_\theta(\boldsymbol{z}_k)\right) \ . \tag{18}
\end{aligned}$$

Since $\boldsymbol{y}$ and $\boldsymbol{z}$ are assumed to be marginally independent, *i.e.*, $p_\theta(\boldsymbol{y};\boldsymbol{z}) = p_\theta(\boldsymbol{y})p_\theta(\boldsymbol{z})$, then

$$\begin{aligned}
&I_\theta(\boldsymbol{x};\boldsymbol{y}) + \sum_{k=1}^{K_1} I_\theta(\boldsymbol{x};\boldsymbol{z}_k) - D_{\mathrm{KL}}\left(p_\theta(\boldsymbol{y},\boldsymbol{z})||p_\theta(\boldsymbol{y})\Pi_{k=1}^{K_1}p_\theta(\boldsymbol{z}_k)\right)\\
&= I_\theta(\boldsymbol{x};\boldsymbol{y}) + \sum_{k=1}^{K_1} I_\theta(\boldsymbol{x};\boldsymbol{z}_k) - D_{\mathrm{KL}}\left(p_\theta(\boldsymbol{z})||\Pi_{k=1}^{K_1}p_\theta(\boldsymbol{z}_k)\right) \ . \tag{19}
\end{aligned}$$

**Proof of proposition 1**

*Proof.* We start with computing the expectation of $\boldsymbol{z}_k$:

$$\mathbb{E}_\theta\left[\boldsymbol{z}_k\right] = \int_{\mathcal{Z}_k} \boldsymbol{z}_k \int_{\mathcal{X}} p_\theta(\boldsymbol{z}_k|\boldsymbol{x})p(\boldsymbol{x})d\boldsymbol{x}d\boldsymbol{z}_k = \int_{\mathcal{X}} p(\boldsymbol{x}) \int_{\mathcal{Z}_k} \boldsymbol{z}_k p_\theta(\boldsymbol{z}_k|\boldsymbol{x})d\boldsymbol{z}_k d\boldsymbol{x}$$

$$= \int_{\mathcal{X}} p(\boldsymbol{x})\mu_k(\boldsymbol{x})d\boldsymbol{x} = \mathbb{E}_{\boldsymbol{x}}\left[\mu_k(\boldsymbol{x})\right] . \qquad (20)$$

Then the variance of $\boldsymbol{z}_k$ followed as:

$$\mathrm{Var}_\theta\left[\boldsymbol{z}_k\right] = \int_{\mathcal{Z}_k} \boldsymbol{z}_k^2 \int_{\mathcal{X}} p_\theta(\boldsymbol{z}_k|\boldsymbol{x})p(\boldsymbol{x})d\boldsymbol{x}d\boldsymbol{z}_k - \mathbb{E}_{\boldsymbol{x}}\left[\mu_k(\boldsymbol{x})\right]^2$$

$$= \int_{\mathcal{X}} p(\boldsymbol{x}) \int_{\mathcal{Z}_k} \boldsymbol{z}_k^2 p_\theta(\boldsymbol{z}_k|\boldsymbol{x})d\boldsymbol{z}_k d\boldsymbol{x} - \mathbb{E}_{\boldsymbol{x}}\left[\mu_k(\boldsymbol{x})\right]^2$$

$$= \int_{\mathcal{X}} p(\boldsymbol{x})\left[\sigma_k^2(\boldsymbol{x}) + \mu_k(\boldsymbol{x})^2\right]d\boldsymbol{x} - \mathbb{E}_{\boldsymbol{x}}\left[\mu_k(\boldsymbol{x})\right]^2$$

$$= \mathbb{E}_{\boldsymbol{x}}\left[\sigma_k^2(\boldsymbol{x})\right] + \mathrm{Var}_{\boldsymbol{x}}\left[\mu_k(\boldsymbol{x})\right] . \qquad (21)$$

Note that

$$I_\theta(\boldsymbol{x};\boldsymbol{z}_k) = H_\theta(\boldsymbol{z}_k) - H_\theta(\boldsymbol{z}_k|\boldsymbol{x}) , \qquad (22)$$

for which we have the following,

$$H_\theta(\boldsymbol{z}_k|\boldsymbol{x}) = -\int_{\mathcal{X}} p(\boldsymbol{x}) \int_{\mathcal{Z}_k} p_\theta(\boldsymbol{z}_k|\boldsymbol{x})\log p_\theta(\boldsymbol{z}_k|\boldsymbol{x})d\boldsymbol{z}d\boldsymbol{x}$$

$$= \frac{1}{2}\int_{\mathcal{X}} p(\boldsymbol{x})\log\left(2\pi e\sigma_k^2(\boldsymbol{x})\right)d\boldsymbol{x}$$

$$= \frac{1}{2}\left(\log(2\pi e) + \mathbb{E}_{\boldsymbol{x}}\left[\log\sigma_k^2(\boldsymbol{x})\right]\right) . \qquad (23)$$

For the entropy of $\boldsymbol{z}_k$, we leverage the fact that $H_\theta(\boldsymbol{z}_k)$ is upper bounded by the entropy of a Gaussian distributed random variable with the same mean and variance, that is

$$H_\theta(\boldsymbol{z}_k) \leq \frac{1}{2}\left(\log 2\pi e + \log\left(\mathbb{E}_{\boldsymbol{x}}\left[\sigma_k^2(\boldsymbol{x})\right] + \mathrm{Var}_{\boldsymbol{x}}\left[\mu_k(\boldsymbol{x})\right]\right)\right) \qquad (24)$$

Substituting Eqs (23) & (24) into Eq (22) completes the proof. $\qquad \square$

**Proof of proposition 2**

*Proof.* Let $\widehat{p}_\theta(\boldsymbol{y}) = \frac{1}{N}\sum_{n=1}^N p_\theta(\boldsymbol{y}|\boldsymbol{x}_n)$ denote the Monte Carlo estimator of the true probability $p_\theta(\boldsymbol{y}) = \int_{\mathcal{X}} p(\boldsymbol{x})p_\theta(\boldsymbol{y}|\boldsymbol{x})d\boldsymbol{x} = \mathbb{E}_{\boldsymbol{x}}\left[p_\theta(\boldsymbol{y}|\boldsymbol{x})\right]$. Note that $p_\theta(\boldsymbol{y}|\boldsymbol{x}) \in [0,1]$ for all $\boldsymbol{x} \in \mathcal{X}$, then applying the Hoeffding's inequality for bounded random variables [Theorem 2.2.6, (Vershynin, 2018)] yields,

$$\mathbb{P}\left(|\widehat{p}_\theta(\boldsymbol{y}) - p_\theta(\boldsymbol{y})| \geq t\right) = \mathbb{P}\left(\left|\frac{1}{N}\sum_{n=1}^N p_\theta(\boldsymbol{y}|\boldsymbol{x}_n) - \mathbb{E}_{\boldsymbol{x}}\left[p_\theta(\boldsymbol{y}|\boldsymbol{x})\right]\right| \geq t\right) \leq 2\exp\left(-2Nt^2\right) \quad (25)$$

Let $\delta' = 2\exp\left(-2Nt^2\right)$, it then follows,

$$\mathbb{P}\left(|\widehat{p}_\theta(\boldsymbol{y}) - p_\theta(\boldsymbol{y})| < \sqrt{\frac{\log(2/\delta')}{2N}}\right) \geq 1 - \delta' \qquad (26)$$

Given Eq (26), we first establish the concentration results of the entropy $H_{\widehat{p}_\theta}(\boldsymbol{y})$ with respect to the empirical distribution $\widehat{p}_\theta(\boldsymbol{y})$. Assume For all $y \in \mathcal{C}$, we have $p_\theta(y), \widehat{p}_\theta(y)$ bounded below by $1/(CK_2)$ for some fixed constant $C > 1$. This assumption is practical since the distributions of true

data and predicted data are approximately uniform and therefore $p_\theta(y), \widehat{p}_\theta(y) \approx 1/K_2$ for all $y \in \mathcal{C}$. Consider the function $t \log t$, with derivative $1 + \log t \in [1 - \log CK_2, 1]$ for $t \in [1/(CK_2), 1]$,

$$|\widehat{p}_\theta(y) \log \widehat{p}_\theta(y) - p_\theta(y) \log p_\theta(y)| = \left| \int_{p_\theta(y)}^{\widehat{p}_\theta(y)} (1 + \log t) dt \right|$$

$$\leq \left| \int_{p_\theta(y)}^{\widehat{p}_\theta(y)} |1 + \log t| dt \right| \leq \left| \int_{p_\theta(y)}^{\widehat{p}_\theta(y)} \max\{\log CK_2 - 1, 1\} dt \right|$$

$$\leq \max\{\log CK_2 - 1, 1\} |\widehat{p}_\theta(y) - p_\theta(y)| \tag{27}$$

Summing over $\mathcal{C}$ gives

$$\left| \widehat{H}_\theta(\boldsymbol{y}) - H_\theta(\boldsymbol{y}) \right| \leq K_2 \max\{\log CK_2 - 1, 1\} |\widehat{p}_\theta(y) - p_\theta(y)| . \tag{28}$$

Let $\delta = K_2 \delta'$, then Eq (26) together with Eq (28) yield the following,

$$\mathbb{P}\left( \left| \widehat{H}_\theta(\boldsymbol{y}) - H_\theta(\boldsymbol{y}) \right| < K_2 \max\{\log CK_2 - 1, 1\} \sqrt{\frac{\log(2K_2/\delta)}{2N}} \right) \geq 1 - \delta \tag{29}$$

Next we are going to bound the divergence between $\widehat{H}_\theta(\boldsymbol{y}|\boldsymbol{x})$ and $H_\theta(\boldsymbol{y}|\boldsymbol{x})$ which are defined as,

$$\widehat{H}_\theta(\boldsymbol{y}|\boldsymbol{x}) = -\frac{1}{N} \sum_{n=1}^{N} \sum_{\boldsymbol{y}} p_\theta(\boldsymbol{y}|\boldsymbol{x}_n) \log p_\theta(\boldsymbol{y}|\boldsymbol{x}_n),$$

$$H_\theta(\boldsymbol{y}|\boldsymbol{x}) = -\int_{\boldsymbol{x} \in \mathcal{X}} \sum_{\boldsymbol{y}} p_\theta(\boldsymbol{y}|\boldsymbol{x}) \log p_\theta(\boldsymbol{y}|\boldsymbol{x}) .$$

Note that $h \log h \in [-1/e, 0]$ for all $h \in [0, 1]$, then again applying [Theorem 2.2.6, (Vershynin, 2018)] yields,

$$\mathbb{P}\left( \left| \frac{1}{N} \sum_{n=1}^{N} p_\theta(\boldsymbol{y}|\boldsymbol{x}_n) \log p_\theta(\boldsymbol{y}|\boldsymbol{x}_n) - \mathbb{E}_{p(\boldsymbol{x})} [p_\theta(\boldsymbol{y}|\boldsymbol{x}) \log p_\theta(\boldsymbol{y}|\boldsymbol{x})] \right| < t \right) \leq 2 \exp\left( -2t^2 e^2 N \right) \tag{30}$$

Following the similar arguments as before, let $\delta' = 2 \exp\left( -2t^2 e^2 N \right)$, then

$$\mathbb{P}\left( \left| \frac{1}{N} \sum_{n=1}^{N} p_\theta(\boldsymbol{y}|\boldsymbol{x}_n) \log p_\theta(\boldsymbol{y}|\boldsymbol{x}_n) - \mathbb{E}_{p(\boldsymbol{x})} [p_\theta(\boldsymbol{y}|\boldsymbol{x}) \log p_\theta(\boldsymbol{y}|\boldsymbol{x})] \right| < \sqrt{\frac{e^2 \log(2/\delta')}{2N}} \right) \leq \delta' \tag{31}$$

Now let $\delta = K_2 \delta'$, then applying the union bound we have

$$|\widehat{H}_\theta(\boldsymbol{y}|\boldsymbol{x}) - H_\theta(\boldsymbol{y}|\boldsymbol{x})| \leq \sum_{\boldsymbol{y} \in \mathcal{C}} \left| \frac{1}{N} \sum_{n=1}^{N} p_\theta(\boldsymbol{y}|\boldsymbol{x}_n) \log p_\theta(\boldsymbol{y}|\boldsymbol{x}_n) - \mathbb{E}_{p(\boldsymbol{x})} [p_\theta(\boldsymbol{y}|\boldsymbol{x}) \log p_\theta(\boldsymbol{y}|\boldsymbol{x})] \right|$$

$$\leq K_2 \sqrt{\frac{e^2 \log(2K_2/\delta)}{2N}} \tag{32}$$

hold with probability $1 - \delta$.

Conclude from Eqs (29) & (32), we have

$$\left| I_\theta(\boldsymbol{x}; \boldsymbol{y}) - \widehat{I}_\theta(\boldsymbol{x}; \boldsymbol{y}) \right| \leq \left| H_\theta(\boldsymbol{y}) - \widehat{H}_\theta(\boldsymbol{y}) \right| + \left| H_\theta(\boldsymbol{y}|\boldsymbol{x}) - \widehat{H}_\theta(\boldsymbol{y}|\boldsymbol{x}) \right|$$

$$= K_2 \left( \max\{\log CK_2 - 1, 1\} + e \right) \sqrt{\frac{\log(2K_2/\delta)}{N}} . \tag{33}$$

hold with probability at least $1 - 2\delta$. □

## B    APPROXIMATION OF THE MARGINAL DISTRIBUTION

Computing the marginal distributions of the continuous representations $\boldsymbol{z}$ and $\boldsymbol{z}_k$ requires the entire dataset, *e.g.,* $p_\theta(\boldsymbol{z}) = \int_\mathcal{X} p_\theta(\boldsymbol{z}, \boldsymbol{x}) d\boldsymbol{x} \approx \frac{1}{N} \sum_{i=1}^N p_\theta(\boldsymbol{z}|\boldsymbol{x}^{(i)})$. To scale up our method to large datasets, we propose to estimate based on the minibatch data, *e.g.,* $p_\theta(\boldsymbol{z}) \approx \frac{1}{B} \sum_{i=1}^B p_\theta(\boldsymbol{z}|\boldsymbol{x}^{(i)})$.

Now consider the entropy $\mathrm{H}(\boldsymbol{z})$ of $\boldsymbol{z}$, which we approximate in the following way,

$$\mathrm{H}(\boldsymbol{z}) = \mathbb{E}_{\boldsymbol{z}}[\log p(\boldsymbol{z})] \approx \frac{1}{B} \sum_{i=1}^B \log p(\boldsymbol{z}^{(i)}) = \frac{1}{B} \sum_{i=1}^B \log \frac{1}{B} \sum_{j=1}^B p_\theta(\boldsymbol{z}^{(i)}|\boldsymbol{x}^{(j)}) . \tag{34}$$

We estimate the integral of $\boldsymbol{z}$ by sampling $\boldsymbol{z} \sim p_\theta(\boldsymbol{z}|\boldsymbol{x}_i)$ and perform the Monte Carlo approximation. Although we minimize the unbiased estimator of the lower bound of the KL divergence, the term inside the logarithm is a summation of probability densities of Gaussians. In particular, we record the distribution of the variances output by our encoder and observe that the mean of the variances of the Gaussians is bounded between 0.2 and 2, which implies that the values of probability densities do not range in a large scale. Since logarithm is locally affine, we argue that our bound in (34) is tight. Other quantities involved in our objective function (10) are estimated in a similar fashion.

## C    CONNECTIONS TO VAE

In VAE, they assume a generative model specified by a stochastic decoder $p_\theta(\boldsymbol{x}|\boldsymbol{z})$, taking the continuous representation as an example, and seek an encoder $q_\phi(\boldsymbol{z}|\boldsymbol{x})$ as a variational approximation of the true posterior $p_\theta(\boldsymbol{z}|\boldsymbol{x})$. The model is fitted by maximizing the evidence lower bound (ELBO) of the marginal likelihood,

$$\mathbb{E}_{\boldsymbol{x}}\left[\log p_\theta(\boldsymbol{x})\right] \geq \mathcal{L}(\boldsymbol{x}, \theta, \phi) = \mathbb{E}_{q_\phi(\boldsymbol{z}|\boldsymbol{x})}\left[\log p_\theta(\boldsymbol{x}|\boldsymbol{z})\right] - \mathbb{E}_{\boldsymbol{x}}\left[D_{\mathrm{KL}}\left(q_\phi(\boldsymbol{z}|\boldsymbol{x})||r(\boldsymbol{z})\right)\right] . \tag{35}$$

Here the KL divergence term can be further decomposed as (Hoffman and Johnson, 2016),

$$\mathbb{E}_{\boldsymbol{x}}\left[D_{\mathrm{KL}}\left(q_\phi(\boldsymbol{z}|\boldsymbol{x})||r(\boldsymbol{z})\right)\right] = I_\theta(\boldsymbol{x}; \boldsymbol{z}) + \mathbb{E}_{\boldsymbol{x}}\left[D_{\mathrm{KL}}\left(q_\phi(\boldsymbol{z})||r(\boldsymbol{z})\right)\right] . \tag{36}$$

That is, minimizing the KL divergence also penalizes the mutual information $I_\theta(\boldsymbol{x}; \boldsymbol{z})$, thus reduces the amount of information $\boldsymbol{z}$ has about $\boldsymbol{x}$. This can make the inference task $q_\phi(\boldsymbol{z}|\boldsymbol{x})$ hard and lead to poor reconstructions of $\boldsymbol{x}$ as well. Many recent efforts have been focused on resolving this problem by revising ELBO. Although approaches differ, it can be summarized as either dropping the mutual information term in Eq (36), or encouraging statistical independence across the dimensions of $\boldsymbol{z}$ by increasing the penalty on the total correlation term extracted from the KL divergence $D_{\mathrm{KL}}\left(q_\phi(\boldsymbol{z})||r(\boldsymbol{z})\right)$ with respect to $q_\phi(\boldsymbol{z})$. However, these approaches either result in an invalid lower bound for the VAE objective, or cannot avoid minimizing the mutual information $I_\theta(\boldsymbol{x}; \boldsymbol{z})$ between the representation and the data.

In contrast, IMAE starts with a stochastic encoder $p_\theta(\boldsymbol{z}|\boldsymbol{x})$ and aims at maximizing the mutual information between the data $\boldsymbol{x}$ and the representations $\boldsymbol{z}$ from the very beginning. By following the constraints which are naturally implied by the objective in order to avoid degenerated solutions, IMAE targets at both informative and statistical independent representations. On the other hand, in IMAE the decoder $q_\phi(\boldsymbol{x}|\boldsymbol{z})$ serves as a variational approximation to the true posterior $p_\theta(\boldsymbol{x}|\boldsymbol{z})$. As we will show in Section 4, being able to learn more interpretable representations allows IMAE to reconstruct and generate data with better quality.

## D VAT STABILIZES THE LEARNING OF CATEGORICAL REPRESENTATIONS

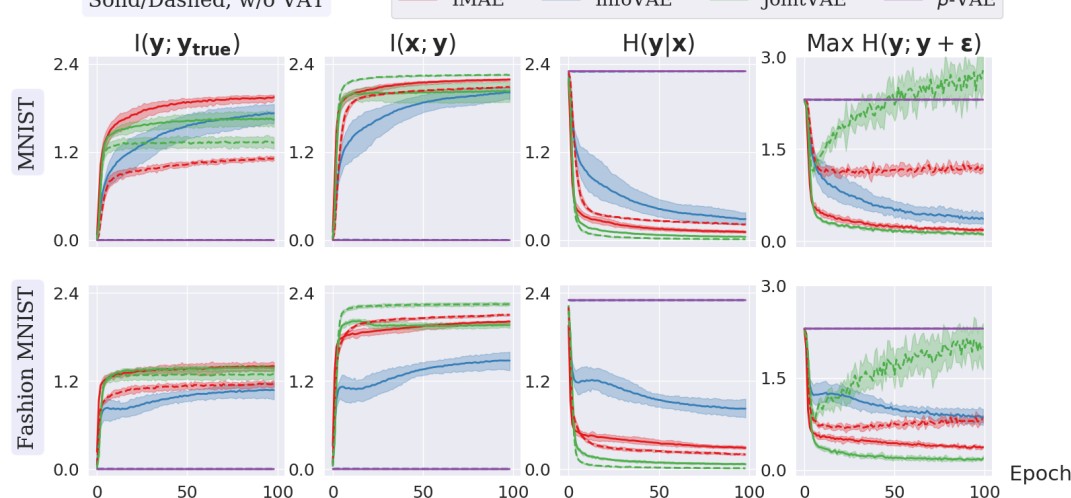

Figure 6: Prevent over confidence predictions by encouraging local smoothness

## E DISENTANGLEMENT QUALITY WITH RESPECT TO BOTH CONTINUOUS AND DISCRETE REPRESENTATIONS ON 2D SHAPES

See figure 7.

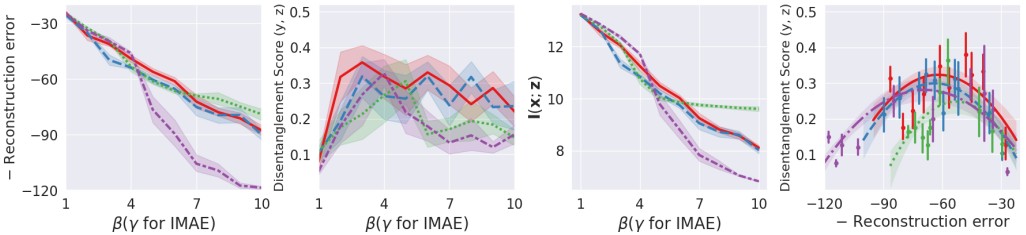

(a) IMAE performs well regarding the disentanglement score vs. decoding quality trade-off, especially in the region of interest where both decoding quality and informativeness of representations are fairly good.

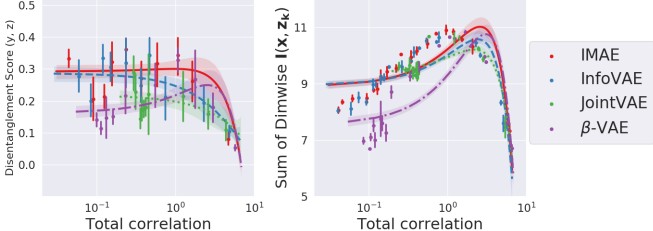

(b) Negative correlation between total correlation and disentanglement score. It also implies that the disentanglement score tends to decrease along with the total correlation if using even larger $\beta$, due to the diminishing informativeness of representation factors. In the extreme case, both total correlation and disentanglement score can degrade to zero.

Figure 7: **Disentanglement comparison on dSprites with respect to both $y$ and $z$.** The results are reported by training each method with $\beta \in [1, 10]$, and we set $\beta = \gamma/2$ with $\gamma \in [1, 10]$ for IMAE. For each $\beta$ value, every method is trained over 8 random initializations. Shade regions indicate the 80% confidence intervals.

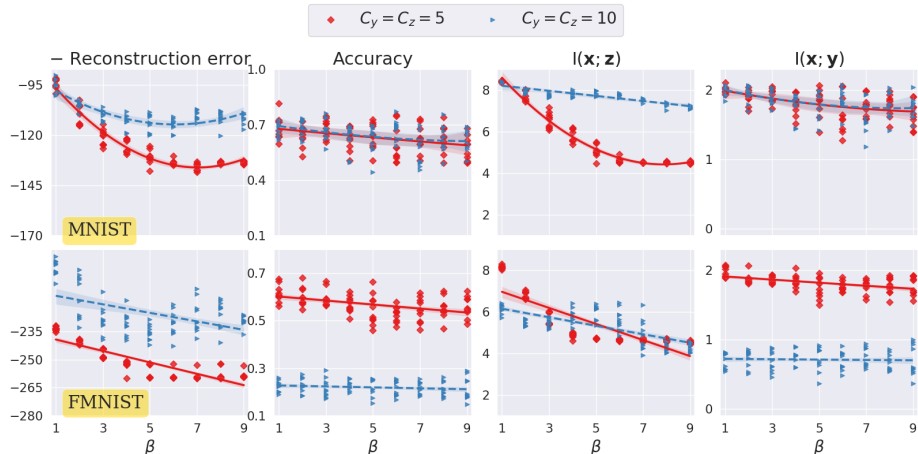

Figure 8: JointVAE with different sets of target vlues $(C_y, C_z)$. For each $\beta$ value, we train JointVAE with 10 different random seeds. We augment JointVAE with VAT.

## F    MORE RESULTS ON JOINTVAE

See figure 8.

## G    EXPERIMENTAL SETTINGS

Table 1: Encoder and Decoder architecture for MNIST and Fashion MNIST.

| Encoder | Decoder |
| --- | --- |
| Input vectorized $28 \times 28$ grayscale image | Input $\boldsymbol{y} \in \mathbb{R}^{10}$ and $\boldsymbol{z} \in \mathbb{R}^{10}$ |
| FC. 500 BatchNorm ReLU | FC. 500 ReLU |
| FC. $2 \times 500$ BatchNorm ReLU | FC. 500 ReLU |
| FC. 20 $(\mu_{\boldsymbol{z}}, \log \sigma_{\boldsymbol{z}})$ + 10 $(p_{\boldsymbol{y}})$ | FC. $28 \times 28$ Sigmoid |

Table 2: Encoder and Decoder architecture for dSprites.

| Encoder | Decoder |
| --- | --- |
| Input vectorized $64 \times 64$ grayscale image | Input $\boldsymbol{y} \in \mathbb{R}^{3}$ and $\boldsymbol{z} \in \mathbb{R}^{10}$ |
| FC. 1200 ReLU | FC. 1200 ReLU |
| FC. 1200 ReLU | FC. 1200 ReLU |
| FC. $2 \times 1200$ ReLU | FC. 1200 ReLU |
| FC. 20 $(\mu_{\boldsymbol{z}}, \log \sigma_{\boldsymbol{z}})$ + 3 $(p_{\boldsymbol{y}})$ | FC. $28 \times 28$ Sigmoid |

**Training procedure:**

- **MNIST & Fashion MNIST:** We use momentum to train all models. The initial learning rate is set as $1e$-3, and we decay the learning rate by $0.98$ every epoch.
- **dSprites:** We use Adam to train all models. The learning rate is set as $1e$-3.

