# OpenReview forum: "INFORMATION MAXIMIZATION AUTO-ENCODING"
_ICLR.cc/2019/Conference_

### Official Review · AnonReviewer1 · 2018-10-24
**Not Compelling.**

**Rating:** 4
**Confidence:** 4

**Review:**

This paper proposes an objective function for auto-encoding they
call information maximizing auto encoding (IMAE).  To set the stage
for my review I will start with the following "classical" formulation
of auto-encoding as the minimization of the following where we are
training models for P(z|x) and P(x|z).

beta H(z) + E_{x,z sim P(z|x)} -log P(x|z) (1)

Here H(z) is defined by drawing x from the population and then drawing
z from P(z|x).  This is equivalent to classical rate-distortion coding
when P(x|z) is an isotropic Gaussian in which case -log P(x|z) is just
the L2 distortion between x and its reconstruction.  The parameter
beta controls the trade-off between the compression rate and the L2
distortion.

This paper replaces minimizing (1) with maximizing

beta I(x,z) + E_{x,z sim P(z|x)} log P(x|z) (2)

This is equivalent to replacing H(z) in (1) by -I(x,z).  But (2)
admits a trivial solution of z=x.  To prevent the trivial solution this
paper proposes to regularize P(z) toward a
desired distribution Q(z) and replacing I(x,z) with KL(P(z),Q(z))
by minimizing

beta KL(P(z),Q(z)) + E_{x,z sim P(z|x)} - log P(x|z) (3)

The paper contains an argument that this replacement is reasonable
when Q(z) and P(z|x) are both Gaussian with diagonal covariances.  I
did not verify that argument but in any case it seems (3) is better than (2).
For beta large (3) forces P(z) = Q(z) which fixes H(z) and the a-priori value
H(Q).  The regularization probably has other benefits.

But these suggestions are fairly simple and any real assessment of their
value must be done empirically.  The papers experiments with MNIST
seem insufficient for this.

---

> ### Author Response · Authors · 2018-11-27
> **Clarifying the objective as well as the motivations for the proposed approach**
>
> We'd like to thank the reviewer for the comments.  Below we further clarify our objective and address your concern regrading the numerical results.
>
> 1) First of all, we want to point out the reviewer could misunderstand our objective. To be more specific,  proposition 1 is regrading  a single continuous representation factor z_k (which is a scalar variable), therefore the trivial solution z = x suggested by the reviewer does not exists.  In other words, the degenerate solution we seek to avoid is respect to each dimension of the continuous representation instead of z itself.  Moreover, in this paper, we focus on learning low dimensional yet interpretable representations of the data.
>
> Using the setting you provided above, suppose z in R^K and the conditional distribution P(z|x) is factorial (which is a typical assumption is the VAE literature), we first decompose I(x, z) as the following:
>
>                          I(x, z) = sum_k=1^K   I(x, z_k) - KL(P(Z) || product_k=1^K P(Z_k))
>
> where the first term of RHS quantifies the informativeness of each dimension z_k, and the second term is often referred as the "total correlation" of z which achieves the minimum (zero) if all dimensions of z are independent of each other.  That is the mutual information I(x, z) inherently involves two keys terms that quantify the informativeness of each representation factor and the statistical dependence between these factors. We then propose to maximize informativeness of each (scalar) representation factor z_k while simultaneously encourage statistical independence across latent factors by minimizing the "total correlation" term. By doing so, we are expected to learn informative yet more disentangled representations (see figure 5).
>
> For each scalar representation factor z_k, we (mathematically) show in section 3.2 that the trivial solution of maximizing I(x, z_k)  can be obtained by severely fragmenting the latent space. To be more specifically, the mutual information I(x, z_k) can be trivially maximized by mapping each data sample to a deterministic value of z_k, while dispersing the different z_k values associated with different data samples within a dramatically large space. This can results in discontinuity of the latent representations, which is not desired.  A natural resolution for this problem is to restrict the variance the z_k to be a reasonable value, with which we propose to push the the marginal distribution of P(z_k) towards a gaussian distribution so as to achieve the upper bound of I(x, z_k) (eq (4)) in proposition 1.
>
>
> 2) We also want to emphasize that, we propose a framework to learn a hybrid discrete-continuous representations of the data.  We seek to learn semantically meaningful discrete representations while maintaining disentanglement of the continuous representations that capture the variations shared across categories. Unsupervised joint learning of disentangled continuous and discrete representations is a challenging problem due to the lack of prior for semantic awareness and other inherent difficulties that arise in learning discrete representations.
>
> To the best of our knowledge, our work is, apart from JointVAE, the only framework for jointly learning discrete and continuous representations in a completely unsupervised setting in the VAE literature.
>
> 3) We update the paper by considering more challenging dataset and incorporating more quantitative evaluations regarding the trade-off between interpretability of representations and decoding quality.  Please refer to "summarization of revision" provided above.

---

> > ### Comment · AnonReviewer1 · 2018-11-27
> > **Empirical evaluation still insufficient**
> >
> > I have looked at the revision and also simply looked at the paper more carefully.  My earlier review was somewhat careless because, in any case, the empirical evaluation is quite weak.  Although I believe that a stronger empirical evaluation is required, the fundamental issues being addressed are important and it is nice to get things clarified.
> >
> > Your only parameters are the encoder and decoder parameters.  These parameters do not support computing marginal distributions on continuous latent variables.  Consider the term KL(P_Theta(z_k),r(z_k)) appearing in equation (6) and ultimately appearing in the overall loss function.  This term equals
> >
> > E_{z_k} log (P_Theta(z_k)/r(z_k))
> >
> > We can sample z_k by sampling x and sampling z_k from P_Theta(z|x).  Since z_k is one dimensional we could try to model P_Theta(z_k) as a one dimensional Gaussian by empirically measuring its mean and variance.  But this would not be a good model if P_Theta(z|x) is trending toward delta distributions.  While the problem of P_Theta(z|x) drifting toward delta distributions is discussed, the method proposed does not seem to address the problem.  A simple fix is to require that P_Theta(z|x) has a minimum variance.  This possibility is mentioned but not formally placed in the objective.
> >
> > This problem is much more serious in the total variation term in the final objective (10).  To optimize this term we need to be able to assign a joint marginal probability P_Theta(z) to a particular sample of the vector z. ???

---

> > > ### Author Response · Authors · 2018-11-28
> > > **Estimating the marginal distributions**
> > >
> > > Thank you for bring this question up for discussion! We address your concerns one by one below.
> > >
> > > (I) -- We first show the marginals we need to estimate and how we estimate them. Note that r(z) is a factorial gaussian, then
> > >
> > >           KL[p(z) || prod_k p(z_k)] +  sum_k KL[p(z_k) || r(z_k)]  = KL[p(z) || r(z)]      (1)
> > >
> > > Hence our objective is equivalent to
> > >
> > >            L_IMAE = reconstruction error + beta * L(y)  - beta* KL[p(z) || r(z)] + (beta - gamma) * KL[p(z) || prod_k p(z_k)]       (2)
> > >
> > > Therefore, to optimize (2), we need to estimate KL[p(z) || r(z)] and total correlation KL[p(z) || prod_k p(z_k)].  Now we explain how to estimate these two terms.
> > >
> > >     (a)  To estimate KL[p(z) || r(z)], let B denote the batch size
> > >            KL[p(z) || r(z)]   =   E_p(z) log p(z) - E_p(z) log r(z)   \approx   (1/B) * sum_i  log p(z^i)   -   (1/B) * sum_i log r(z^i)     (3)
> > > where z^i is sampled from p_theta(z|x^i).   We still need to approximate log p(z^i) in (3), which can be estimated as the following:
> > >                            log p(z^i)  \approx  log (1/B) * sum_j   p(z^i |x^j)     (4)
> > >
> > >     (b)  Similar arguments for the total correlation term,
> > >                       KL[p(z) || prod_k p(z_k)] = E_p(z) log p(z) - E_p(z) log prod_k p(z_k)   (5)
> > >        We have established the estimator for the first term E_p(z) log p(z), and the second term can be estimated in a similar way, i.e.,
> > >                     E_p(z) log prod_k p(z_k)  =  sum_k  E_p(z) log p(z_k)
> > >                                                             \approx   (1/B) * sum_k sum_i  log p(z_k^i)
> > >                                                             \approx   (1/B) * sum_k  sum_i log  (1/B) * sum_j   p(z_k^i | x_j)
> > > where z_k^i is sampled from p(z | x^i) as before.
> > >
> > >
> > > (II)-- Regarding the concern of small variance for the conditional distribution p(z|x),  we want to point out the following:
> > >
> > >       (a)  Minimizing KL divergence KL[p(z) || r(z)] will drive the variance away from being very small. As you can see in (3) and (4), if the variance of the conditional distribution is very small, then the KL divergence can be very large too. This is also numerically demonstrated in Figure 2 of our paper, where you can see the expectation of sigma_k^2 is not very small across dimensions, even for those informative dimensions the associated E[sigma_k^2] have reasonable values. Figure 1(a) is obtained with beta = 2, E[sigma_k^2] of the informative dimensions can be larger by using large beta values.
> > >
> > >     (b)  We propose to squeeze the marginal distribution p(z_k) within a gaussian distribution with finite mean so as to avoid the degenerate solution where p(z_k |x) can be delta distribution. By doing so, we can also achieve the maximum of I(x; z_k) among all possible solutions with the same variance of z_k, i.e. Var(z_k) being the same. Please refer to section 3.2 for more discussion.
> > >           Another advantage of pushing p(z_k) instead of the conditional distribution p(z_k|x) towards a gaussian distribution is that, pushing p(z_k |x^i) of each data sample i towards the same gaussian distribution can result in undesired overlap between different p(z_k | x), this would cause serious reconstruction problem and loss of informativeness in z_k as well. In the extreme case, when p(z_k|x) converged to the target gaussian distribution, the representation z_k carries zero information about the data (see Fig 1 in the WAE paper https://openreview.net/pdf?id=HkL7n1-0b ).  Although, by requiring the variance of p(z_k|x) being larger than some prefixed value instead of pushing it towards a gaussian distribution might improve the issue a bit, this will introduce another data dependent hyperparameter to tune. Moreover, with a large set of training data, pushing p(z_k) towards r(z_k) while requiring the variance of the conditional distribution p(z_k|x) being larger than some fixed value can still induce overlap.  For these two reasons, we prefer to not adding such constraint.  By pushing p(z_k) towards a gaussian distribution with reasonable variance, it can balance well between pushing z_k|x^i apart for different samples x^i while maintaining reasonable variance for the conditional distribution.

---

> > > > ### Author Response · Authors · 2018-11-28
> > > > **typos got corrected**
> > > >
> > > > We corrected some typos in the above response we just posted. Let us know if you have any questions. Thank you!

---

> > > > ### Comment · AnonReviewer1 · 2018-11-30
> > > > **Question not addressed**
> > > >
> > > > I don't think your response addresses my question.  How are you computing p(z_k)?

---

> > > > > ### Author Response · Authors · 2018-12-01
> > > > > **Response to reviewer 1**
> > > > >
> > > > > In our paper, we assumed p(z|x) is factorial, hence p(z_k) can be estimated in the same way used to estimate p(z).
> > > > >
> > > > > To explicitly estimate p(z) or p(z_k), we need to pass the whole dataset, i.e., p(z) = sum_i p(z|x^i) / N. In this paper, we estimate p(z) using minibatch data with size B ( set to be 512 to 2048 for all methods). Based on this, we approximate all key quantities (the relevant expectations in KL divergence terms in (2)) using MC estimation, which are explicitly calculated via (3) (4) and (5).
> > > > >
> > > > > Let us know if you have any further questions. Thank you!

---

### Official Review · AnonReviewer3 · 2018-10-31
**Principled framework for auto-encoding**

**Rating:** 6
**Confidence:** 4

**Review:**

* This paper proposed a principled framework for auto-encoding through information maximization. A novel contribution of this paper is to introduce a hybrid continuous-discrete representation. The authors also related this approach with other related work such as \beta-VAE and info-VAE, putting their work in context. Empirical results show that the learned representation has better trade-off among interpretability and decoding quality.

* It seems a little strange to me to incorporate the VAT regularization to the IMAE framework in Section 4.2, as this is not included in the overall objective in Equation (10) and earlier analysis (Proposition 1 and 2). Will the conclusions in Proposition 1 and 2 change accordingly due to the inclusion of VAT regularization?

* The paper states that IMAE has better trade-off among interpretability and decoding quality. But it is still unclear how a user can choose a good trade-off according to different applications. More discussion along this direction would be helpful.

* I guess the L(y) term in Equation (10) is from Equation (9), but this is not stated explicitly in the paper.

---

> ### Author Response · Authors · 2018-11-27
> **Update the revision by providing more quantitative evaluations regrading the interpretability vs. decoding quality over various datasets**
>
> We sincerely thank the reviewer for the positive feedback and the constructive comments/questions. In order to address the main concerns, we incorporate more quantitative comparisons and provide more comprehensive numerical results to evaluate IMAE, which we summarized above. Below are our answers for your questions.
>
> 1)*  It seems a little strange to me to incorporate the VAT regularization to the IMAE framework in Section 4.2, as this is not included in the overall objective in Equation (10) and earlier analysis (Proposition 1 and 2). Will the conclusions in Proposition 1 and 2 change accordingly due to the inclusion of VAT regularization?
>
>     VAT is proposed to resolve the inherent difficulty of learning interpretable discrete representations using neural network. As we mentioned at the beginning of section 4.2,  the high capacity of neural network makes it easy to learn a non-smooth function p(y|x) that can abruptly change its predictions without guaranteeing similar data samples will be mapped to similar y.  VAT is proposed as a regularization to encourage local smoothness of the conditional distribution p(y|x) for discrete representations.
>
>     In our experimental results, we found that using VAT are significantly helpful for learning interpretable discrete representations for all methods considered in this paper except betaVAE. More interpretable continuous representations can be obtained when the method is capable of learning discrete representations that match the true categorical information of data better,  since less overlap between the manifolds of each category is induced. This in turn can better help the continuous representations to capture the variation (feature) information shared over different categories while simultaneously reducing the possibility for the continuous representations to encode the nuisance information between separated manifolds of each category.
>
>     Propositions 1&2 are provided without considering VAT. As we discussed above, VAT is proposed as a regularization term. Based on our discussions above, we hypothesize that similar statements can still be true under mild assumption ( e.g., the categorical data are comparatively separated and there does exist common feature information shared over different categories.  Since VAT is incorporated to promote the local smoothness of p(y|x), this shouldn't influence proposition 1 where the I(x, y) is defined w.r.t the global information between x & y.  Proposition 2 is true in general regardless of y.   (Intuitively, including VAT can help continuous representations to better focus on learning feature information shared across categories, since VAT helps learn more interpretable y.
>
> 2) * The paper states that IMAE has better trade-off among interpretability and decoding quality. But it is still unclear how a user can choose a good trade-off according to different applications. More discussion along this direction would be helpful.
>
>     In the revision, we comprehensively evaluate IMAE against the other three methods on various datasets.   For each dataset, we train each method with a wide range of hyperparameter values.  The corresponding results are summarized in Figure 3 (MNIST and Fashion MNIST) and Figures 5&7 (dSprites).
>
>     As shown in Figures 3,  IMAE consistently outperforms the other three methods in terms of learning more interpretable (accurate) discrete representations over a wide range of hyperparameter values, while simultaneously achieving comparatively better decoding quality and more informative representations. This is further demonstrated in Figure 5 where we evaluate IMAE on a more challenging dataset (dSprites) and quantitatively evaluate the disentanglement vs. decoding quality trade-off.  Still, IMAE performs better regarding the disentanglement score vs. reconstruction trade-off over a wide range of beta, gamma values.
>
>     Moreover, as demonstrated in both figure 3 and figure 5, in the region of interest where both the reconstruction error and the informativeness of representations are fairly good,  IMAE achieves a much better reconstruction error vs. interpretability trade-off. We attribute this to the effects of 1) maximizing mutual information I(X, Y) is capable of learning more interpretable discrete representations that match the natural labels of data better;  2) explicitly promoting statistically independent continuous latent factors by minimizing the total correlation term in our objective. By using comparatively large weight on the total correlation term (we set gamma = 2*beta in this paper.), we are able to achieve better disentanglement without sacrificing the decoding quality too much.
>
>
> 3)* I guess the L(y) term in Equation (10) is from Equation (9), but this is not stated explicitly in the paper.
>
>     Thank you for capturing the typo, which we corrected in the updated version.

---

### Official Review · AnonReviewer2 · 2018-11-01
**Idea is promising and the derivation of the loss is informative but the evaluation seems insufficient.**

**Rating:** 5
**Confidence:** 5

**Review:**

Summary: the paper proposes a method for unsupervised disentangling of both discrete and continuous factors of variation in image data. It uses an autoencoder learned by optimising an additive loss composed of Mutual Information (MI) I(x;y,z) between the image x and the discrete+cts latents (y,z) and the reconstruction error. The mutual information is shown to decompose into I(x,y), I(x,z) and TC(y;z), and the I(x,z) is treated in a different manner to I(x,y). With Gaussian p(z|x), and it is shown that I(x,z_k) is maximal when p(z_k) is Gaussian. So KL(p(z_k)||N(0,1)) is optimised in lieu of optimising I(x,z), and I(x,y) (and TC(y;z)) is optimised by using mini-batch estimates of marginal distributions of y (and z). The paper claims improved disentangling of discrete and continuous latents compared to methods such as JointVAE and InfoVAE.

Pros:
- The derivation of the loss shows a nice link between Mutual information and total correlation in the latents.
- It is a sensible idea to treat the MI terms of the discrete latents differently to the continuous latents
- The mathematical and quantitative analysis of MI and its relation to decoder means and variances are informative.

Cons:
- There is not enough quantitative comparison of the quality of disentanglement across the different methods. The only values for this are the accuracy scores of the discrete factor, but for the continuous latents there are only qualitative latent traversals of single models, and I think these aren’t enough for comparing different disentangling methods - this is too prone to cherry-picking. I think it’s definitely necessary to report some metrics for disentangling that are averaged across multiple models trained with different random seeds. I understand that there are no ground truth cts factors for Mnist/FashionMnist, but this makes me think that a dataset such as dSprites (aka 2D Shapes) where the factors are known and has a mix of discrete and continuous factors would have been more suitable. Here you can use various metrics proposed in Eastwood et al, Kim et al, Chen et al for a quantitative comparison of the disentangled representations.
- In figure 4, it says beta=lamda=5 for all models. Shouldn’t you be doing a hyperparameter sweep for each model and choose the best value of hyperparameters for each? It could well be that beta=5 works best for IMAE but other values of beta/lambda can work better for the other models.
- When comparing against JointVAE, the authors point out that the accuracy for JointVAE is worse than that of IMAE, a sign of overfitting. You also say that VAT helps maintain local smoothness so as to prevent overfitting. Then shouldn’t you also be comparing against JointVAE + VAT? Looking at Appendix D, it seems like VAT makes a big difference in terms of I(y;y_true), so I’m guessing it will also have a big impact on the accuracy. Thus JointVAE + VAT might beat IMAE in terms of accuracy as well, at which point it will be hard to argue that IMAE is superior in learning the discrete factor.
- In the first paragraph of Section 4, the authors claim results on CelebA, but these are missing from the paper. Testing the approach on datasets more complex than (Fashion)Mnist would have been desirable.
- There aren’t any latent traversals for the discrete latents - this would be a useful visualisation to complement the accuracy plots in Figure 3.

Qs and comments:
- It’s not clear why posterior approximation quality (used as a starting point for motivating the loss) is an important quantity for disentangling.
- I see that the upper bound to I(x;z_k) in (4) and the objective in (6) have the same optimum at p(z_k) being Gaussian, but it’s not clear that increasing one leads to increasing the other. Using (6) to replace (4) seems to require further justification, whether it be mathematical or empirical.
- In proposition 2, I’m sceptical as to how meaningful the derived bound is, especially when you set N to be the size of the minibatch (B) in practice. It also seems that for small delta (i.e. to ensure high probability on the bound) and large K_2 (less restrictive conditions on p(y) and \hat{p}(y)), the bound can be quite big.
- \mathcal{L}_theta(y) in equation (10) hasn’t been introduced yet.
- The z dimension indices in the latent traversal plots of Figure 2 don’t seem to match the x-axis of the left figure. It’s not clear which are the estimates of I(x;z_k) for k=8,3,1 in the figure.

---

> ### Author Response · Authors · 2018-11-27
> **Response to Reviewer 2 (part 1): updated the numerical results by considering more complext dataset and incorporating more quantitative comparisons**
>
> We sincerely thank the reviewer for the thoughtful comments and suggestions. In order to address the main concerns, we updated the numerical results by considering a more complex dataset and incorporating the suggested quantitative evaluations. Below, we start by addressing your concerns one by one.
>
> 1) * There is not enough quantitative comparison of the quality of disentanglement across the different methods.
>
>     We fully agree that a quantitative comparison of the disentanglement quality regrading both continuous and discrete representations significantly improves the paper. We provide a quantitative comparison in terms of the disentanglement quality vs. reconstruction error trade-off on dSprites. The corresponding results are summarized in Figures 5&7, where we train each method over a wide range of hyperparameter values, for each value we train over 8 random seeds.
>
>     We found that, IMAE consistently performs better in terms of achieving better disentanglement quality vs. reconstruction error trade-off over a wide range of beta, gamma values. We attribute this to the effect of explicitly promoting statistical independent continuous latent factors in our objective. Compared to InfoVAE, by using comparatively large weight on the total correlation terms (we set gamma=2*beta in this paper), we are able to achieve better disentanglement quality without sacrificing the decoding quality too much. This allows us to obtain better disentanglement vs. reconstruction error trade-off, especially in the region where both the informativeness of latent representations and the decoding quality are fairly good.
>
>     Although JointVAE attains better decoding quality as well as more informative (overall) representations with large beta values, the associated disentanglement quality is poor. We suspect that simply pushing the upper bound the mutual information towards a target value does not explicitly encourage disentanglement across the representation factors.
>
>
> 2)*  Shouldn’t you be doing a hyperparameter sweep for each model and choose the best value of hyperparameters for each?
>
>     Thank you for the comments! For the updated numerical results, we do sweep over a wide range of hyperparameter values and for each value we run every method over 10 random seeds (8 for dSprites due to the limited computational resource, will increases it to 10 or 15 later).
>
> 3)* Looking at Appendix D, it seems like VAT makes a big difference in terms of I(y;y_true), so I’m guessing it will also have a big impact on the accuracy. Thus JointVAE + VAT might beat IMAE in terms of accuracy as well, at which point it will be hard to argue that IMAE is superior in learning the discrete factor.
>
>     In the initial version, we actually augmented all models with VAT in the numerical section,  which can significantly improve all methods except betaVAE. The comparison between the results obtained by using (solid) and not using (dashed) VAT is provided in appendix. We do apologize if we didn't make it clear in the original submission. Same for the updated numerical results, we augment all models with VAT.
>
>     We also provide one more result (Figure 8 in Appendix F) for JointVAE by running it with different target values C_y&C_z. Although JointVAE can achieve better reconstruction error by using larger target values C_y&C_z, the corresponding disentanglement / interpretability of representation factors can be very poor (Figs 5&7).
>
>
> 4)* In the first paragraph of Section 4, the authors claim results on CelebA, but these are missing from the paper. Testing the approach on datasets more complex than (Fashion)Mnist would have been desirable.
>
>     We were not able to do conduct experiments on celebA due to the time constraint and the limited computational resource.  As for the updated version, we do quantitatively evaluate our approach on more challenging dataset (dSprites) against the other three approaches.  However, we are not able to conduct comprehensive experiments on celebA for the same reason. We do apologize for that, we will incorporate the corresponding results (hopefully also 3D chairs) for the final version.
>
> 5)* There aren’t any latent traversals for the discrete latents - this would be a useful visualisation to complement the accuracy plots in Figure 3.
>
>      Thank you for the suggestion! We incorporate the associated results in both Figure 2 and Figure 4.
>
>
> We answered your questions in a separate note.

---

> ### Author Response · Authors · 2018-11-27
> **Response to Reviewer 2 (part 2): regarding your questions and comments:**
>
> 1)--We seek to learn interpretable representations together with a decoding/generative model, where informative representations can then be leveraged to generate high fidelity data. The relationship between I(x; (y,z)) and KL(p(x|y,z) || q(x|y,z)) can be interpreted according to the following:
>                        KL(p(x|y,z) || q(x|y,z)) = CrossEntropy(p(x|y,z), q(x|y,z))  - Entropy(p(x|y,z))   (*)
>                       I(x; y,z) = Entropy(x) - Entropy(p(x|y,z)  (**)
>       (*) and  (**) implies the following:
>             a) Since Entropy(x) is independent of the optimization procedure of (**), maximizing  I(x; y,z) decreases Entropy(p(x|y,z)).  On the other hand, Minimizing KL will push CrossEntropy  towards Entropy(p(x|y,z)).  Hence, jointly optimizing I(x; y,z) and KL can yield informative representations  as well as good posterior approximation quality (both KL divergence and CrossEntropy(p(x|y,z), q(x|y,z)) are small).
>             b) A good balance can be obtained by setting the weight on I(x; y,z) larger than 1, since if the weight is one,
>                   I(x; y,z) - KL(p(x; y,z) || q(x; y,z)) = Entropy(x) -  CrossEntropy(p(x|y,z), q(x|y,z)) (***)
>                That is the model degenerate to a plain auto-encoder, optimizing (***) is equivalent to simply optimizing CrossEntropy(p(x|y,z), q(x|y,z)) (the reconstruction error). Therefore, by setting the weight larger than 1, we can simultaneously attain good posterior approximation quality and informative representations with desired distributions.
>
> 2)-- You are right, without any assumptions, increasing (6) does not necessary increase (4). However, it is true if we restrict the variance of $\zb_k$ to be a fixed value, given which increasing (6) does lead to the increase of (4). Specifically,  the proposed objective (6) can be justified by the following:
>         a) with the same amount of variance in z_k (Var(z_k) is fixed),   I(x, z_k) is maximized if p(z_k) is gaussian;
>         b) As we discussed in section 3.2,  I(x, z_k) can be trivially maximized by pushing pushing the condtional means mu_k(x) being extremely farway from each other while diminishing the conditional variance sigma_k(x) to zeros. This can result in a severely fragmented latent space where the distribution of z_k are discontinuous.  To remedy this issue, restricting the variance of the latent representation (Var(z_k)) to be some finite value is a natural resolution.  Given this, squeezing the distribution of z_k within the domain a gaussian distribution with finite variance achieves the maximal mutual information (upper bound in (4)) among all possible solutions with the same variance of z_k.
>
> 3)--Thank you for bring this question up for discussion, for which we want to point out the   following:
>         a) The bound in proposition 2 depends on log(delta) and log (C ), therefore the required number of samples won't increase dramatically by pursuing high probability bound with less restrictive assumptions on p(y) and \hat{p}(y).
>         b) The required number of samples N is on the order of K_2^2 for large K_2.  However, unsupervised learning of categorical representation with large number of categories itself is very challenging. A possible resolution is to learn the representations over a multiple-stage procedure, at each stage we learn a small number of categories within the single parent category, the theoretical guarantee still valid.
>         c) Proposition 2 is proved by considering the worst case, hence the estimation error can be much better in practice.
>
> 4)--Thank you for capturing this, which we corrected in the revision ( see Eq (9)).
>
> 5)-- We do apologize for the confusion in the initial figure, where the indices of the left plot correspond with the sorted values of I(x; z_k), while k=8,3,1 denoting the indices of z_k without sorting them. To avoid the confusion, we use the indices w.r.t the sorted values for all four plots in the revision.

---

> > ### Comment · AnonReviewer2 · 2018-11-30
> > **Response to Author Response**
> >
> > - The draft changed substantially from the original submission, especially the experimental section. As per the guidelines of the reviewing process on the conference webpage, it would not be fair to increase the score based on these drastic changes.
> > - Regarding discrete latent traversals, it seems like you’ve used different z’s across the rows of Figures 2b,c,d and for 2e,f it’s not clear whether you’re keeping z fixed across the latent traversal of y. Similar point holds for Figure 4a. It would be infromative to keep the z's fixed across the discrete latent traversals to show good disentangling.
> > - I can’t see how if you fix the variance of z_k, then increasing (6) leads to increasing (4). Can this be shown mathematically? Also don’t you learn the variance of z_k in practice? You seem to denote it as sigma_k(x), implying that it is learned.

---

> > > ### Author Response · Authors · 2018-12-01
> > > **Response to reviewer 2**
> > >
> > > The reason for us to update the numerical section substantially is that we agree with the reviewers, the numerical evaluation in our initial submission is insufficient and cannot provide fairly informative comparisons between IMAE and the other models. We want to add the numerical results (on dSprites) suggested by you, since we do think it can improve the paper and provides more insights. However, we do understand if  you prefer to not change the score due those changes.
> > >
> > > As for your questions:
> > > 1) We do appreciate the suggestion! We will add it in a later version.  In our current plots,  for each row, the initial z is the average value of all z's (from testing data) whose labels are predicted as y by IMAE. Therefore, we are performing latent traverse based on the averaged z associated with each learnt category.
> > >
> > > 2) Var(z_k) does not equal to sigma_k(x), since sigma_k(x) is the variance of the conditional distribution  p_\theta(z|x) and Var(z_k) is the variance of the marginal z_k.  It can be proven that (Eq (21) in appendix),
> > >        Var(z_k) = E_x [sigma_k^2(x)] + Var[mu_k(x)]  (*)
> > > where mu and sigma denote the mean and variance of p_\theta(z|x) respectively, and p_\theta(z|x) is assumed to be gaussian.  Now assume Var(z_k) is fixed, then
> > >            I_\theta(x; z) <= 0.5 log Var(z_k)  -  E_x [log sigma_k^2(x)]     (**)
> > > Var(z_k) is calculated in (*), the upper bound in (**) is only attained when z_k is gaussian.  ((*) and (**) are proved in the proof of proposition 1 in appendix).
> > >
> > > Therefore, we push p(z_k) towards a scaled normal distribution r(z_k) ~ N(0, alpha) with alpha being some prefixed value for Var(z_k),  so as to achieve the maximal I_\theta(x, z_k) among all possible solutions with Var(z_k) = alpha.
> > >
> > > We discussed in section 3.2 (also in the previous response to you) about the reasons for restricting Var(z_k) being a (fixed) finite value.
> > >
> > > Let us know if you have any questions.

---

### Author Response · Authors · 2018-11-27
**Summarization of the revision**

We sincerely thank all reviewers for the thoughtful comments and suggestions. To address your main concerns, we updated our numerical results by considering a more challenging dataset and incorporating more quantitative comparisons. Here is a summarization of the revision:

1) We updated the results on MNIST and Fashion MNIST by sweeping over a range of hyperparameter values for all methods considered in the paper. For each value, we ran each method over 10 random seeds, and the results are summarized in Figure 3.

As you can see, IMAE consistently outperforms the other three methods in terms of learning more interpretable discrete representations over a wide range of hyperparameter values, while simultaneously achieving comparatively better decoding quality and more informative representations. Moreover, in the region of interest where both decoding quality and informativeness of representations are fairly good,  IMAE achieves a much better decoding quality vs. interpretability trade-off. (This is also true when we consider more complex dataset (dSprites), please refer to Figures 5&7.)

On the other hand, being capable of learning more interpretable discrete representations that match the ground truth of the categorical information better will also induce more interpretable continuous representations followed by better decoding quality, since there is less overlap between the manifolds corresponding with each category. (see figure 4).


2)  We added another quantitative comparison regrading the disentanglement vs. decoding quality trade-off. We trained all four methods on 2D shapes (dSprites) over a wide range of hyperparameter values, and evaluated the disentanglement quality by using the metric proposed by (Chen et al 2018).  The associated results are provided in Figures 5&7, where for each hyperparameter value we ran over 8 different random seeds. (Currently, we are only able to provide the results over 8 random initializations due to the limited computational resource we have, we will update the results with 10-15 random seeds later.)

As shown in Figures 5&7, IMAE performs well regarding the disentanglement score vs. decoding quality trade-off which is especially better in the in the region of interest where the decoding quality as well as the informativeness of latent representations are fairly good. We attribute this to the effect of explicitly seeking statistically independent latent factors by minimizing the total correlation term in our objective. In other words, by putting comparatively larger weight on the total correlation term (we set gamma = 2*beta in this paper), we are able to achieve better disentanglement along with good decoding quality.

---

> ### Author Response · Authors · 2018-11-27
> **Regarding your questions and comments:**
>
> 1) We seek to learn interpretable representations together with a decoding/generative model, where informative representations can then be leveraged to generate high fidelity data. The decoder is included to approximate the posterior distribution of the data given their representations, which together with the learnt distributions of the representations can be used to generate data after training. The relationship between I(x; (y,z)) and KL(p(x|y,z) || q(x|y,z)) can be interpreted according to the following:
>
>                        KL(p(x|y,z) || q(x|y,z)) = CrossEntropy(p(x|y,z), q(x|y,z))  - Entropy(p(x|y,z))   (*)
>
>      (*) implies the following:
>         b) Note that I(x; y,z) = Entropy(x) - Entropy(p(x|y,z) where Entropy(x) is independent of the optimization procedure, hence maximizing  I(x; y,z) decreases the value of Entropy(p(x|y,z)).  On the other hand, Minimizing KL(p(x|y,z) || q(x|y,z)) will push CrossEntropy(p(x|y,z), q(x|y,z))  towards Entropy(p(x|y,z)).  Therefore, jointly optimizing I(x; y,z) and KL(p(x|y,z) || q(x|y,z)) can simultaneously yield informative representations  as well as good posterior approximation quality.
>
>       c) A good balance can be obtained by putting comparatively large weight (>1) on I(x; y,z), since if the weight is one, then
>                    I(x; y,z) - KL(p(x; y,z) || q(x; y,z)) = Entropy(x) -  CrossEntropy(p(x|y,z), q(x|y,z)) (**)
>      That is the model degenerate to a plain auto-encoder, optimizing (**) is equivalent to simply optimizing CrossEntropy(p(x|y,z), q(x|y,z)) (the reconstruction error).
>        In contrast, by using large weight on I(x; y, z), we can simultaneously attain informative representations with desired distributions as well as good posterior approximation quality (see (b)).
>
>
> 2): You are right, increasing (6) does not necessary increase (4) without any assumptions. It's true if we restrict the variance of $\zb_k$ to be a fixed value, given which increasing (6) does lead to the increase of (4).
>
>     The proposition of (6) can be justified by the following:
>         a) with the same amount of variance in z_k (Var(z_k) is a fixed value),  the mutual information I(x, z_k) is maximized if p(z_k) is gaussian;
>         b) It's reasonable (even necessary) to restrict the variance of the latent representation factor to be some finite value so as to avoid degenerate solution. Moreover,  I(x, z_k) can be trivially maximized by pushing pushing the condtional means mu_k(x) being extremely farway from each other while simultaneously diminishing sigma_k(x) to zeros. This can result in a severely fragmented latent space where the distribution of z_k are discontinuous.
>            Therefore, restrict the variance of the latent representation (Var(z_k)) to be some reasonable finite value is a natural resolution to avoid undesired representations.  Given this, squeezing the distribution of z_k within the domain a gaussian distribution with finite variance achieves the maximal mutual information (upper bound in (4)) among all possible solutions with the same variance of z_k.
>
> 3)-Thank you for bring this question up for discussion, for which we want to point out the   following:
>         a): The bound in proposition 2 depends on log(delta) and log (C ), therefore the required number of samples won't increase dramatically by pursuing high probability bound with less restrictive assumptions on p(y) and \hat{p}(y).
>         b): The required number of samples N is on the order of K_2^2 for large K_2. It does require a large batch size when we consider large K_2. However, unsupervised learning of categorical representation with large number of categories itself is very challenging. A possible resolution is to learn the representations over a multiple-stage procedure, at each stage we learn a small number of categories within the single parent category. By doing so, we are still able to learn the categorical representation for data with a large number of categories by using the proposed method with theoretical guarantee.
>
> 4)-Thank you for capturing this, which we corrected in the revision ( see Eq (9)).
>
> 5)- We do apologize for the confusion induced by the initial submission, where the indices of the left plot actually index the sorted values of I(x; z_k), while k=8,3,1 denotes the indices of z_k without sorting them. To avoid the confusion, we use the indices of the sorted values for all four plots in the revision.

---

### Comment · Area_Chair1 · 2018-11-30
**confusion notation and definition**

Dear authors,

I appreciate that you took the time to write a detailed response, which, however, does not seem to sufficiently convince the reviewers about the merit of your paper.

I went through your paper a couple of times by myself, and I got quite confused from the very beginning:

First, From Appendix C, it seems that the authors know very well about the convention of defining p_{\theta}(x|z) as the decoder and q_{\phi}(z|x) as the encoder in VAE, but why in the main body of the paper you choose to use p_{\theta}(z|x) as the encoder and q_{\phi}(x|z) as the decoder? Is there particular reason for this inconsistency and using a new set of notation?

Second, since x denotes the data and z denotes the latent variable, why the paper calls p(x|z) as the posterior?? Isn't that p(x|z) is the conditional likelihood, p(z) is the prior,  p(x)=\int p(x|z) p(z) dz is the marginal likelihood, and p(z|x) = p(x|z)p(z)/p(x) is the posterior?

Am I missing some important definitions to justify these unconventional notation and definition?

Thanks,
AC

---

> ### Author Response · Authors · 2018-12-01
> **Response to AC**
>
> Dear AC,
>
> Thank you for spending time reading the paper! We do apologize for the confusion caused by the notations used in our paper. The reason for us to use different notations is that the proposed IMAE approach is significantly different from VAE, or say IMAE is the reversed VAE. To be more specific (we focus on z for better clarification)
>
>     (1) In IMAE, given data x, we start with an probabilistic encoder seek to map it into its representation z, or say we seek to learn the distribution of representation z via a stochastic function p_\theta(z|x) from the very beginning.  In this setting, the joint distribution is modeled as p_\theta(x, z) = p(x) p_\theta(z|x) and the posterior is therefore p_\theta(x|z).  In contrast, in VAE, the situation is reversed, where it starts from a generative model (decoder) p(x|z), the joint distribution is modeled as p(x,z) = p(z)p(x|z) correspondingly and the posterior is p(z|x).
>
>         In summary, IMAE starts with a representation learning model, while VAE starts with a generative latent model, which are the opposite of each other.
>
>    (2) Given (1), p_\theta(z) = \int p(x) p(z|x) is the marginal distribution of z in IMAE, instead of being the prior in VAE.  Although, IMAE does push the marginal distribution p_\theta(z) towards a gaussian prior r(z), the reason is to avoid degenerate solution while simultaneously maximizing I_\theta(x; z) . Please see section 3.2 and our response to reviewer 2 (will be posted soon) for details.
>
>    (3) As a high level summary, IMAE starts from an encoding model, and seeks to maximize the mutual information between the data x and its representation z. A decoder is included to approximate the posterior of the data given its representation, by leveraging the learnt informative representation, better decoding quality is attained. Moreover,  we show that the information maximization objective inherently introduce a balance between informativeness of each representation factor z_k and statistical independence between them.
>
> Let us know if you have any questions.
>
> PS:  We have been hesitant since we first set those notations, since the new notations can reflect the significant difference between IMAE and VAE well, but it can cause confusion too. We are considering changing them back due to confusion induced there.

---

> > ### Comment · Area_Chair1 · 2018-12-01
> > **The definition of posterior is quite unconventional**
> >
> > Since x is the OBSERVED data and z is the LATENT representation, calling p_\theta(x|z) as the posterior is rather unconventional (if not completely wrong).

---

> > > ### Author Response · Authors · 2018-12-02
> > > **regarding the definition of posterior**
> > >
> > > Thank you for bringing this question up for discussion! Yes, we agree that this is incorrect terminology if we interpret x as the observed data and z as the latent variable. However, an encoder model (e.g. IMAE) and a (generative) latent variable are conceptually different. To be more specific,
> > >
> > > 1) In a latent variable model, like the one used in VAE, the distribution of data x is modeled as p(x) = \int_z p_\theta(x|z)p(z)dz, where the prior p(z) is predefined and the conditional probability p_\theta(x|z) is explicitly modeled. We then estimate the parameter z by maximizing the likelihood function.
> > >
> > > 2) In our IMAE model (and other encoder models that maximize MI [3,4]), z is a stochastic function of x that is explicitly modeled as p_\theta(z|x) through a probabilistic encoder. The model is optimized by maximizing the mutual information I_\theta(x,z) between x and z. In this setting, p_\theta(x|z) can be interpreted as the probability of x given the (observed) representation z, which can be interpreted as the posterior probability. In our model, we assume p(z|x) is gaussian, z is “observed” by interpreting it as the response/output of a gaussian channel in information theory. To be more specific, z = mu + epsilon * sigma can be interpreted as adding scaled gaussian noise “sigma*epsilon” to the deterministic encoder mean “mu” (see section 4.2 in [5]).
> > >
> > > The difference is induced by the fundamental conceptual difference between theses two models, and below are some relevant references. We list them here by no means saying the AC’s concern/confusion regarding the posterior is not right. Indeed, we have had an internal debate of how to make this discussion the most clear, accurate, and mathematically correct — We now plan to either write an extensive justification of the overall point of view (even more precise and clear than in [3], which compares to VAE pointing out that the two are opposite) or to revisit the terminology, whatever it takes to make our model clear. We sincerely appreciate you bringing this up.
> > >
> > > [1] Barber, David and Agakov, Felix. The im algorithm: a variational approach to information maximization. In Proceedings of the 16th International Conference on Neural Information Processing Systems, pp. 201–208. 2003.
> > > https://pdfs.semanticscholar.org/f586/4b47b1d848e4426319a8bb28efeeaf55a52a.pdf
> > >
> > > [2] Variational Information Maximization in Stochastic Environments. Felix Agakov. PhD Thesis. http://aivalley.com/Papers/thesis_1sp.pdf
> > >
> > >
> > > [3] Auto-Encoding Total Correlation Explanation. Shuyang Gao, Rob Brekelmans, Greg Ver Steeg, Aram Galstyan. https://arxiv.org/pdf/1802.05822.pdf
> > >
> > >
> > > [4] Discovering structure in high-dimensional data through correlation explanation. Ver Steeg, Greg, and Aram Galstyan. In Advances in Neural Information Processing Systems, pp. 577-585. 2014.
> > >
> > > [5] Understanding disentangling in β-VAE. Christopher P. Burgess and Irina Higgins et al. https://arxiv.org/pdf/1804.03599.pdf

---

### Meta-Review · Area_Chair1 · 2018-12-11
**A principled modeling framework hindered by inadequate experiments and confusion notation**

**Confidence:** 5
**Recommendation:** Reject

**Metareview:**

The paper proposes a principled modeling framework to train a stochastic auto-encoder that is regularized with mutual information maximization. For unsupervised learning, this auto-encoder produces a hybrid continuous-discrete latent representation. While the authors' response and revision have partially addressed some of the raised concerns on the technical analyses, the experimental evaluations presented in the paper do not appear adequate to justify the advantages of the proposed method over previously proposed ones, and the clarity (in particular, notation) needs further improvement. The proposed framework and techniques are potentially of interest to the machine learning community, but the paper of its current form fells below the acceptance bar. The authors are encouraged to improve the clarify of the paper and provide more convincing experiments (e.g., on high-dimensional datasets beyond MNIST).